# Novel immunotherapeutics against LGR5 to target multiple cancer types

Hung-Chang Chen[1,12,17], Nico Mueller[1,17], Katherine Stott [2], Chrysa Kapeni[1], Eilidh Rivers[2,13], Carolin M Sauer [1], Flavio Beke [1], Stephen J Walsh[3,14], Nicola Ashman[3,15], Louise O'Brien[1], Amir Rafati Fard [2], Arman Ghodsinia[2], Changtai Li[2], Fadwa Joud[1], Olivier Giger[4], Inti Zlobec[5], Ioana Olan[1], Sarah J Aitken [6,7], Matthew Hoare [1], Richard Mair [1], Eva Serrao[1], James D Brenton [1], Alicia Garcia-Gimenez[8], Simon E Richardson[8], Brian Huntly [8], David R Spring [3], Mikkel-Ole Skjoedt [9,10,16], Karsten Skjødt[11], Marc de la Roche [2 ✉] & Maike de la Roche [1 ✉]

## Abstract

**We have developed and validated a highly specific, versatile antibody to the extracellular domain of human LGR5 (α-LGR5). α-LGR5 detects LGR5 overexpression in >90% of colorectal cancer (CRC), hepatocellular carcinoma (HCC) and pre-B-ALL tumour cells and was used to generate an Antibody-Drug Conjugate (α-LGR5-ADC), Bispecific T-cell Engager (α-LGR5-BiTE) and Chimeric Antigen Receptor (α-LGR5-CAR). α-LGR5-ADC was the most effective modality for targeting LGR5+ cancer cells in vitro and demonstrated potent anti-tumour efficacy in a murine model of human NALM6 pre-B-ALL driving tumour attrition to less than 1% of control treatment. α-LGR5-BiTE treatment was less effective in the pre-B-ALL cancer model yet promoted a twofold reduction in tumour burden. α-LGR5-CAR-T cells also showed specific and potent LGR5+ cancer cell killing in vitro and effective tumour targeting with a fourfold decrease in pre-B-ALL tumour burden relative to controls. Taken together, we show that α-LGR5 can not only be used as a research tool and a biomarker but also provides a versatile building block for a highly effective immune therapeutic portfolio targeting a range of LGR5-expressing cancer cells.**

**Keywords** LGR5; Cancer Immunotherapeutics; ADC; BiTE; CAR
**Subject Categories** Cancer; Immunology; Stem Cells & Regenerative Medicine

## Introduction

Leucine-rich repeat-containing G-protein receptor 5 (LGR5) was originally identified as a target gene of oncogenic Wnt signalling in colorectal cancer cells (Van de Wetering et al, 2002). Subsequent work established murine Lgr5 as a pre-eminent molecular marker of stem cells in the intestinal epithelia (Koo and Clevers, 2014), gastric epithelia (Barker et al, 2010), hair follicle (Snippert et al, 2010), foetal mammary gland (Trejo et al, 2017), nephrons in the developing kidney (Barker et al, 2012) the regenerating liver (Huch et al, 2013) and as tumour initiating cells in the small intestinal epithelia (Schepers et al, 2012).

LGR5 has attracted a great deal of therapeutic interest owing to its overexpression in human malignancies such as CRC (reviewed in (Morgan et al, 2018)), HCC (Yamamoto et al, 2003), gastric and ovarian cancers (McClanahan et al, 2006), basal cell carcinoma (Tanese et al, 2008), ER-negative breast cancers (Hagerling et al, 2020), glioblastoma, (Nakata et al, 2013) and certain B-cell malignancies (Cosgun et al, 2017, 2020). It is important to note that the majority of these studies rely on transcript levels as a measure of LGR5 expression rather than protein levels owing to the paucity of suitable, well-validated commercially available antibodies.

Functionally, LGR5 expression in CRC cell lines is required for proliferation, migration, chemosensitivity, colony formation and in vivo transplantation ability (reviewed in (Morgan et al, 2018)). CRC tumour cells are heterogeneous for LGR5 expression; however, in vivo studies using murine-engrafted human primary CRC cells and organoids find that ablation of the LGR5-expressing cellular compartment through treatment with either a specific chemical toxin or chemotherapeutic agent results in tumour

[1]University of Cambridge, Cancer Research UK Cambridge Institute, Robinson Way, Cambridge CB2 0RE, UK. [2]University of Cambridge, Department of Biochemistry, Tennis Court Road, Cambridge CB2 1QW, UK. [3]University of Cambridge, Yusuf Hamied Department of Chemistry, Lensfield Road, Cambridge CB2 1EW, UK. [4]University of Cambridge, Department of Pathology, Tennis Court Road, Cambridge CB2 1QP, UK. [5]Institute of Pathology, University of Bern, Murtenstrasse 31, CH-3008 Bern, Switzerland. [6]University of Cambridge, MRC Toxicology Unit, Tennis Court Road, Cambridge CB2 1QR, UK. [7]Department of Histopathology, Cambridge University Hospitals, NHS Foundation Trust, Main Drive, Cambridge CB2 0QQ, UK. [8]University of Cambridge, Department of Haematology, Puddicombe Way, Cambridge CB2 0AW, UK. [9]Rigshospitalet—University Hospital Copenhagen, Blegdamsvej 9, 2100 Copenhagen, Denmark. [10]Institute of Immunology and Microbiology, University of Copenhagen, Blegdamsvej 3B, 2200 Copenhagen, Denmark. [11]University of Southern Denmark Campusvej 55, Odense M DK-5230, Denmark. [12]Present address: Astra Zeneca, Cambridge, UK. [13]Present address: MRC-University of Glasgow Centre for Virus Research, Glasgow, UK. [14]Present address: Bicycle Therapeutics, Cambridge, UK. [15]Present address: Charles River Laboratories, Saffron Walden, UK. [16]Present address: Novo Nordisk, Måløv, Denmark. [17]These authors contributed equally: Hung-Chang Chen, Nico Mueller. ✉E-mail: mad58@cam.ac.uk; maike.delaroche@cruk.cam.ac.uk

regression (Shimokawa et al, 2017; Kobayashi et al, 2012). Importantly, treatment withdrawal restores tumour proliferation, however regrowth is specific to cells that acquire LGR5 expression. Moreover, one recent study using a mouse model of CRC finds that tumour dissemination and metastatic colonisation is largely a function of cells lacking Lgr5 expression; however, tumour cell proliferation at the primary and metastatic sites obligately required Lgr5 expression (Fumagalli et al, 2020). Thus, prognostic and functional studies of LGR5 expression in CRC and other malignancies indicate that elevated LGR5 expression in tumour cells is crucial for proliferation, marking LGR5 as a promising target for cancer treatment.

Antibody-based immune therapies harness the specificity, efficacy and versatility of antigen-binding moieties for recognition of cell surface proteins expressed by cancer cells. Antibodies have been deployed therapeutically for antibody-dependent cellular cytotoxicity (ADCC), checkpoint inhibitors, antibody-drug conjugates (ADCs) that direct a toxin to cancer cells, or for immune cell re-directing strategies that include bispecific immune cell engagers (e.g. BiTEs) and chimeric antigen receptors (CAR) expression in immune cells (Adams and Weiner, 2005; Fritz and Lenardo, 2019; Drago et al, 2021; Waldman et al, 2020; Slaney et al, 2018). Continuing development of the immunotherapy field will be fuelled by the identification of new molecular targets for malignancies and the development of high-quality antibodies to direct immune system components or cytotoxic agents to solid tumour cancers.

While there is sufficient clinical scope for use of LGR5 antibodies in immune therapies, the lack of suitable antibodies that have undergone robust validation has hampered therapeutic development. Here we report highly specific monoclonal α-LGR5 antibodies that are compatible with a range of experimental and therapeutic applications. Using α-LGR5 for analysing LGR5 protein expression in human tissues and cancers, we establish low to undetectable levels in healthy human tissues, and specific overexpression in CRC, HCC and pre-B-ALL tumours. We describe the development of α-LGR5 as a therapeutic antibody, and its functional validation in targeting LGR5$^+$ CRC and pre-B-ALL cells as an ADC and by directing cytotoxic immune cancer cell killing in the BiTE and CAR modalities. Importantly, we demonstrate robust pre-clinical efficacy of α-LGR5 in all three therapeutic modalities in a murine model of human pre-B-ALL thus supporting continued development of α-LGR5-based immune therapies for all LGR5-expressing cancer types.

# Results

## Generation and validation of antibodies against LGR5

For monoclonal antibody production, we immunised mice with the N-terminal 101 amino acids of the human LGR5 extracellular domain (Fig. EV1A). Coupling of the antigen to diphtheria toxoid was necessary to initiate an effective immune response in immunised mice. B-cell fusions yielded 18 hybridoma clones which were tested against transgenic versions of human and murine LGR family members expressed in HEK293T cells. All LGR family transgenes were flanked by the coding sequence for an N-terminal influenza hemagglutinin (HA) epitope tag and a C-terminal fusion to the vasopressin V2 receptor C-terminal tail (for enhanced protein stability (Snyder et al, 2017)) followed by fusion to eGFP

(Fig. EV1B). We also created a version of human LGR5 with Gly1Ser and Val8Ala substitutions to match the corresponding cynomolgus LGR5 N-terminus (cLGR5); the N-terminal 100 amino acid residues of cLGR5 are otherwise identical to the human LGR5 protein (hLGR5). Western blot analysis of lysates derived from HEK293T cells expressing the various LGR transgenes demonstrated immunoreactivity of the four hybridoma clones (clones 1–4) exclusively to human and cynomolgus LGR5 but not murine Lgr4 and Lgr5 or the closely related human LGR4 and LGR6 (Figs. 1A and EV1C). There was no immunoreactivity of the hybridoma clones to non-transfected HEK293T cells, as these cells do not express LGR5 in the absence of pathway activity.

## All α-LGR5 antibody clones bind a common epitope at the LGR5 N-terminus

The amino acid sequences of the complementary determining regions (CDRs) of the light and heavy chains of clones 1–4 are highly conserved, displaying variation at only four positions (Fig. EV1D). To map the target epitopes, we generated four overlapping fragments encompassing the LGR5 antigenic region, approximately 35 amino acids in length (Fragments 1–4; Fig. EV1A), using the RAD display fusion system (Rossmann et al, 2017). Western blot analysis indicated binding specificity of all four α-LGR5 clones to Fragment 1 (Figs. 1B and EV1E). We further refined the epitope region: all four α-LGR5 clones bound to Fragment 1A (Frag1A) at the N-terminus of the LGR5 protein but not the partially overlapping Frag1B (Figs. EV1A and 1B; EV1E) indicating the N-terminal 15 amino acids of LGR5 contain the epitope. Notably, the sequence of the 15-amino acids epitope diverges substantially from the corresponding region in human LGR4/6 and mouse LGR4/5 but only by two residues from the cLGR5 sequence (Fig. EV1F) explaining the binding specificity of the four α-LGR5 clones.

Binding affinities of α-LGR5 antibody clones for Frag1A/Frag1B were determined by bio-layer interferometry measurements using the Octet platform. The high-affinity and bidentate nature of the two arms of the antibody clones resulted in slow dissociation due to re-binding and the assay required inclusion of 10 μm of a competitor peptide corresponding to the Frag1A sequence. We observed high-affinity binding between the antibody clones and Frag1A with $Kd$ values between 0.76 and 1.4 nM (Table 1). No detectable binding was observed between the antibody clones and Frag1B.

Because of the proximity of the LGR5 Frag1A sequence to amino acids that mediate binding to R-spondin family ligands (Chen et al, 2013), we determined whether antibody binding interfered with Wnt pathway activity in LGR5-expressing cells, measured using the TopFlash assay (Korinek et al, 1997). For these and subsequent studies, we focussed on hybridoma clone 2 (α-LGR5) as its CDR sequences diverged the least amongst the 4 clones. We treated HEK293T cells expressing either hLGR5-eGFP or eGFP with Wnt and R-spondin1, in the presence of greater than 10-fold molar excess of either α-LGR5 (relative to R-spondin1) or murine IgG1 control. There was no significant difference in Wnt/R-spondin1-activated pathway activity between IgG1 or α-LGR5-treated cells (Fig. EV1G) indicating antibody binding to LGR5 does not interfere with R-spondin1 binding.

Taken together, our validation studies establish that α-LGR5 and the other hybridoma clones bind to a common epitope within the

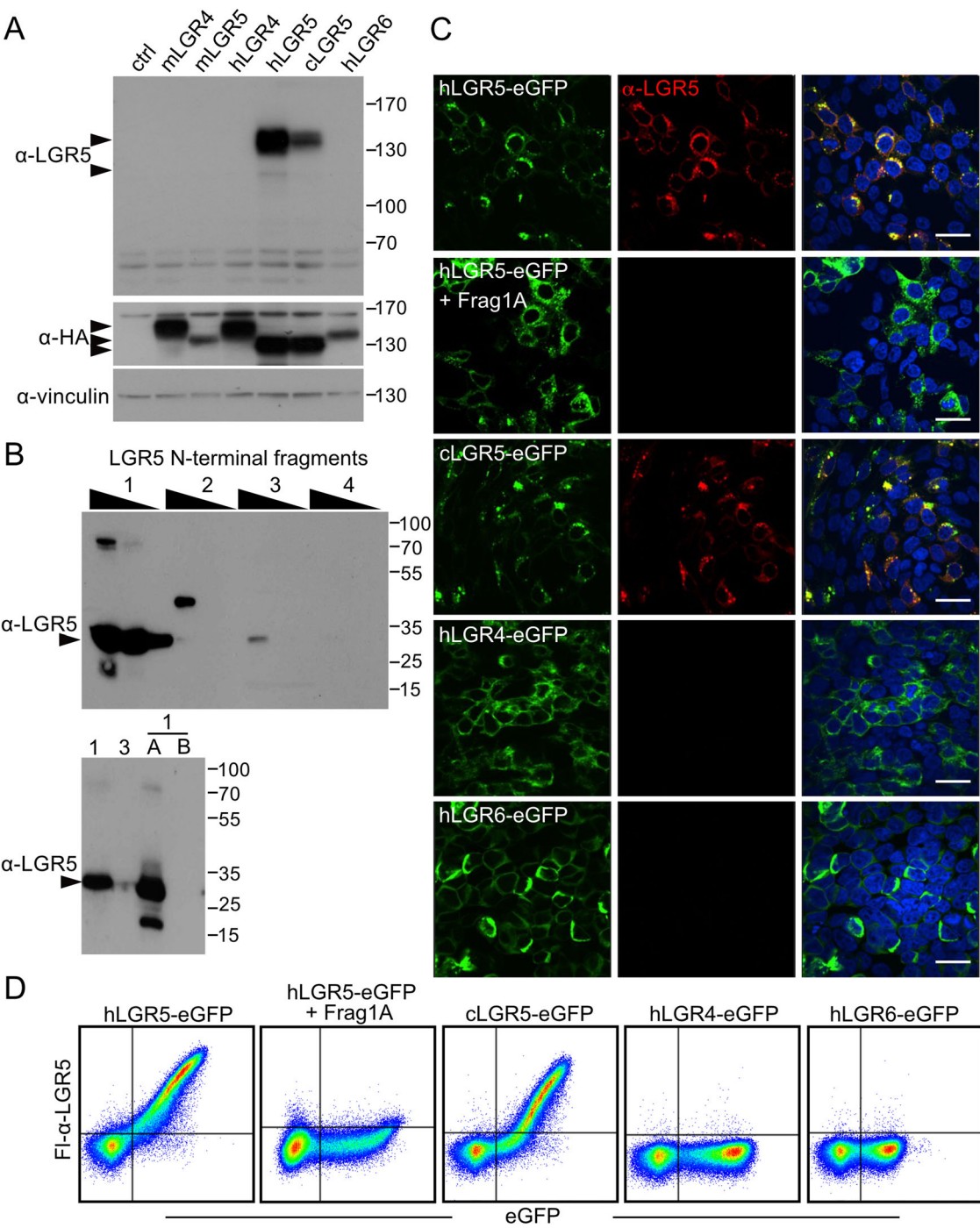

**Figure 1. Validation of novel α-LGR5 antibody clones.**

(A) Western blot analysis of HEK293T lysates expressing LGR family transgenes probed with α-LGR5, and antibodies raised against HA and vinculin. Arrowheads on the side of the top panel indicate the expected position of hLGR5-eGFP (upper) and endogenous LGR5 (lower). Arrowheads on the side of the middle panel indicate the expected sizes of the HA-tagged LGR5 family transgenes. The first lane labelled "ctrl" refers to lysates from HEK293T cells transfected with the parental pEGFP-C2 plasmid. Western blots are representative of three independent experiments. (B) Epitope mapping of α-LGR5 with overlapping fragments from the 101 amino acid antigen sequence (Fig. EV1A) traces the α-LGR5 epitope to Frag1A encompassing the N-terminal 15 amino acids of the mature human LGR5. Western blots are representative of two independent experiments. (C) FL-α-LGR5 detection of LGR5 (red) in HEK293T cells overexpressing eGFP-fused human LGR4-6 transgenes (hLGR4-6) and *cynomolgus* LGR5-eGFP (cLGR5-eGFP; green). In the second row of panels, pre-incubation of FL-α-LGR5 with Frag1A, abolishes the signal for hLGR5-eGFP. Blue, DAPI fluorescence showing nuclei. Scale bars, 10 μm. Images shown are representative of four independent experiments. (D) Flow cytometric analysis of HEK293T cells expressing hLGR family transgenes using FL-α-LGR5 and eGFP for detection. Second panel—pre-incubation of FL-α-LGR5 with Frag1A abrogates the signal detecting hLGR5-eGFP expressing cells. Images are representative of three independent experiments. Source data are available online for this figure.

**Table 1. Binding affinities for murine, humanised and ADC-modified α-LGR5 antibodies.**

| Antibody clone and derivative | Kd (nM) | |
|---|---|---|
| | Frag1A | Frag1B |
| α-LGR5 clone 1 | 0.76 ± 0.01 | ND |
| α-LGR5 clone 2 | 1.1 ± 0.01 | ND |
| α-LGR5 clone 3 | 1.0 ± 0.01 | ND |
| α-LGR5 clone 4 | 1.4 ± 0.01 | ND |
| IgG2 | ND | ND |
| α-LGR5v4 | 2.0 ± 0.02 | ND |
| α-LGR5v6 | ND | ND |
| IgG2-ADC | ND | ND |
| α-LGR5 clone 2-ADC | 2.7 ± 0.03 | ND |
| α-LGR5v4-ADC | 2.0 ± 0.02 | ND |
| α-LGR5v6-ADC | ND | ND |
| α-LGR5 scFv | 0.77 ± 0.3 | ND |

ND not detectable.
Below a 0.1 nm shift in the interference pattern.

N-terminus of human and cLGR5 that does not interfere with binding of R-spondin ligands.

## α-LGR5 antibodies specifically detect human and cynomolgus LGR5 but no other LGR family members in fixed and live cells

To determine antibody specificity in detecting cellular LGR5, we overexpressed the LGR family transgenes in HEK293T cells, fixed the cells in formaldehyde, and probed expression with murine α-LGR5 coupled to Dylight650 fluorophore (Fl-α-LGR5). We did not observe a fluorescence signal from Fl-α-LGR5 in the absence of transgenic expression. However, robust co-localisation of Fl-α-LGR5 and overexpressed hLGR5-eGFP was detected with the Fl-α-LGR5 signal that was abrogated by pre-incubation of the antibody with Frag1A (Fig. 1C). We could also detect the overexpressed cLGR5 transgene with Fl-α-LGR5 in HEK293T cells but not the overexpressed human LGR4, LGR6 or murine LGR4 and LGR5 transgenes (Figs. 1C and EV1H). Flow cytometric analysis of live HEK293T cells overexpressing human, murine, and *cynomolgus* version of LGR family members yielded identical results—hLGR5-eGFP and cLGR5-eGFP overexpressing HEK293T cells were detected by Fl-α-LGR5, but not cells overexpressing the other LGR family transgenes (Figs. 1D and EV1I). Importantly, the signal for cell surface hLGR5-eGFP-expressing cells was abrogated by pre-incubation of Fl-α-LGR5 with Frag1A but not Frag1B (Fig. EV1I).

Taken together, our α-LGR5 antibody specifically identifies overexpressed hLGR5 and cLGR5 by western blot, in fixed cells by immunofluorescence and in live cells by flow cytometry.

## Census of LGR5 expression levels in healthy tissues and cancers

Previous studies have established high LGR5 transcript levels in CRC and some other cancers (Junttila et al, 2015; Gong et al, 2016) raising the possibility of using α-LGR5 as a cancer biomarker or

developing immune-based strategies for therapeutic targeting. We carried out a comprehensive census of LGR5 transcript levels across 33 cancer types using data extracted from the TCGA research network database (https://www.cancer.gov/tcga; Appendix Fig. S1A). We assigned 14 of the cancer types as 'high LGR5 expressors' defined by greater than 70% of component tumour biopsies as harbouring LGR5 expression levels greater than the pan-cancer median (Appendix Fig. S1A). For selected high LGR5 expressors, we compared LGR5 expression to matched healthy tissue. Brain cancer, ovarian cancer, uterine carcinosarcoma, adrenocortical carcinoma and lung squamous carcinoma were excluded from this analysis owing to insufficient data on healthy tissues in the TCGA database. Significant increases in LGR5 transcript expression are apparent in malignancies of the uterine endometrial, stomach, colon and rectum (Appendix Fig. S1B). The exceptions are liver cancer (HCC), where LGR5 expression is depressed relative to healthy tissue, and adrenal gland, oesophageal and pancreatic cancers that are not significantly different in LGR5 expression relative to healthy tissues. Overall, our analyses of pan-cancer LGR5 transcript levels is consistent with a previous study ranking colon and rectum, endometrial carcinoma and ovarian cancers as amongst the highest LGR5 expressors relative to other cancers and to their tissues of origin (Junttila et al, 2015).

We next determined levels of LGR5 protein across a number of healthy tissues and malignancies to evaluate the diagnostic value of our antibody. We used immunofluorescence to score for LGR5 expression using α-LGR5 and a compatible antibody to β-catenin (α-β-catenin) to distinguish epithelial cells. For CRC, we directly compared LGR5 expression levels in three individual biopsies each containing regions of healthy colon epithelia, dysplastic epithelia and cancer. In all three cases, LGR5 expression in the healthy colon epithelia is confined to a small number of β-catenin positive cells, <1% of all β-catenin-expressing epithelial cells appearing as intracellular puncta with no indication of plasma membrane staining (Fig. 2A; Appendix Fig. S1C). Within the dysplastic regions of the tumour, we observe increased cellular LGR5 protein levels in >20% of β-catenin-expressing cells. The adenocarcinoma regions of the tumour display a further increase in LGR5 protein levels (approximately two- to threefold) that are present in >95% of all cancer cells.

We next probed the highly annotated Bern CRC tumour microarray (TMA) with α-LGR5 and α-β-catenin antibodies (Nguyen et al, 2020). Healthy colon epithelia and CRC tumours were scored for levels of LGR5 expression on a scale of 0–3 in β-catenin positive cells: a score of 0 corresponds to the absence of any signal; 1 corresponds to low signal in less than 20% of cells; 2 is either higher signal for LGR5 and/or signal in greater than 20% of cells; and 3 is higher signal for LGR5 in greater than 20% of cells. Of the 213 scored biopsies on the TMA, LGR5 overexpression is apparent in 80% of the tumour samples; however, we observe no significant differences between tumour stages (Fig. 2B) and other phenotypic metrics evaluated for the Bern CRC TMA that include patient age and gender groups, tumour location or the presence of microsatellite instability.

Our LGR5 transcript analyses indicated that LGR5 mRNA expression is significantly equivalent between healthy liver and HCC, directly contrasting previous studies of LGR5 levels in this malignancy (Gong et al, 2016; Junttila et al, 2015). We probed for LGR5 protein levels using a TMA composed of cores from 103

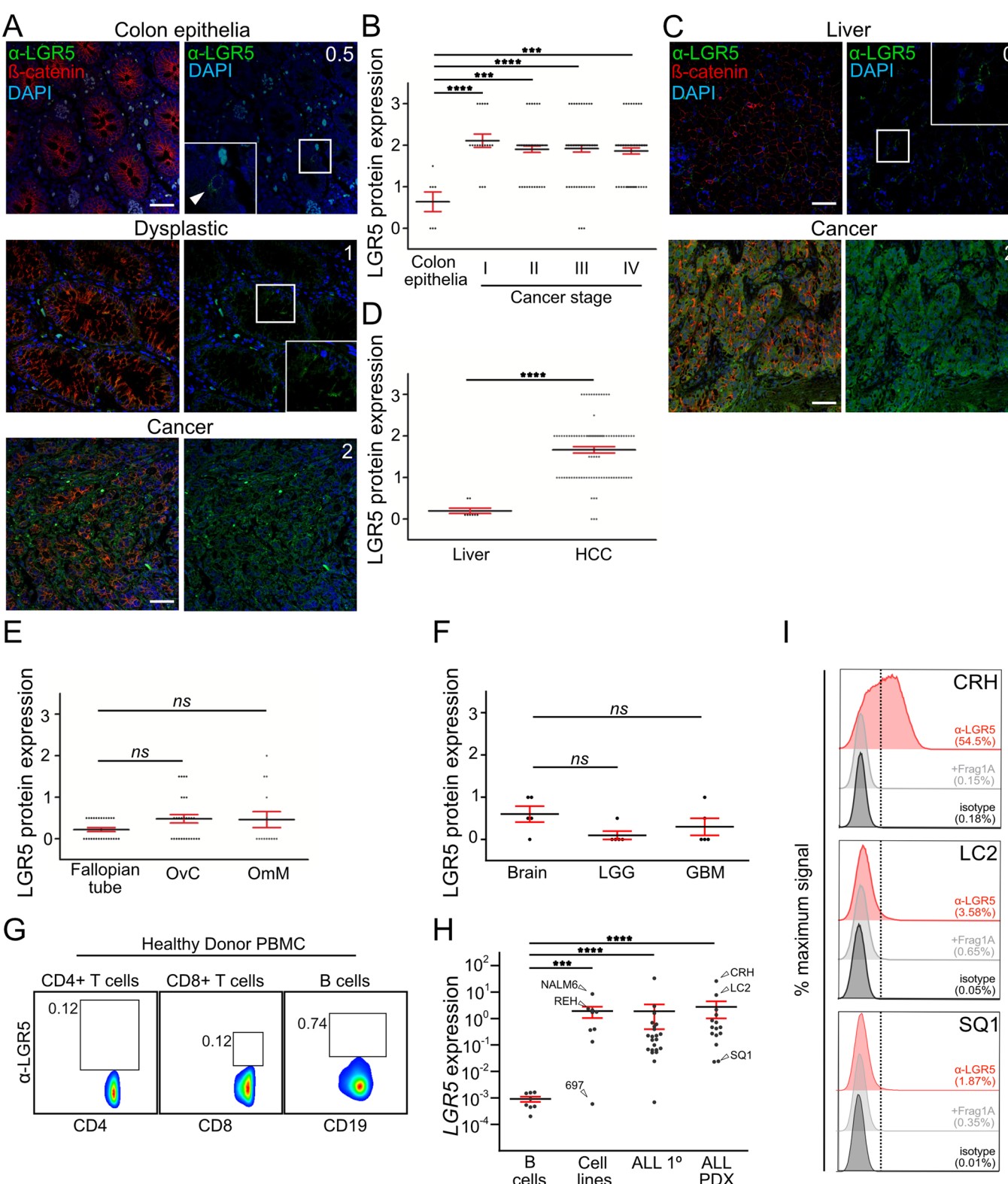

HCC cases, and control healthy liver biopsies taken from five women having undergone resections for inflammatory adenoma (3 cases) or focal nodular hyperplasia (2 cases). In two of the control cases, we detect low LGR5 protein expression in less than 10% of β-catenin positive cells and no expression in the other cases (Fig. 2C; Appendix Fig. S1D). In stark contrast, LGR5 protein levels and number of expressing cells are elevated for greater than 90% of the HCC cases (Fig. 2D; Appendix Fig. S1D). β-catenin is over-expressed in 82% of the cases (Appendix Fig. S1E), and the concordance in overexpression between it and LGR5 was 64%. Of

**Figure 2. Census of healthy tissues and cancers for LGR5 expression levels.**

(A) Sections from a CRC tumour resection showing LGR5 and β-catenin expression in—top panels: normal tissue; middle panels: dysplastic tissue; and lower panels: CRC. Blue, DAPI fluorescence showing nuclei. White numbers in the top right corner of panels (showing LGR5 expression and nuclei staining) correspond to relative levels of LGR5 protein using the scoring criteria applied to all TMAs (see the text). Scale bar, 40 μm. Arrowhead in inset, very rare (<0.1% of all cells) LGR5+ cell in healthy colon epithelium. (B) Relative LGR5 protein expression quantified in healthy colon epithelia and CRC tumour stages I–IV scored. Each dot represents a single-scored biopsy on the Bern 225 biopsy TMA. Biopsies were scored in a single experiment. Date is presented as mean expression, error bars are +/− standard error of the mean (SEM). (C) Immunofluorescence using Fl- α-LGR5 and an antibody to β-catenin for—top panels: a healthy liver sample; and bottom right panels: a sample from the Cambridge HCC TMA. Numbers in white correspond to scored LGR5 expression levels using the criteria for evaluating the TMA. Arrowhead, cell clusters within the tumour with high levels of cortical β-catenin and low levels of LGR5 levels. White numbers represent scored values for relative LGR5 protein expression. Scale bar, 40 μm. (D) Quantitation of LGR5 protein expression levels in eight healthy liver resections (Liver) and biopsies from the Cambridge HCC 105 biopsy TMA (HCC). Biopsies expression levels were scored in a single experiment. Data is presented as mean expression of the biopsies, +/− SEM. (E) Quantitation of LGR5 expression levels in biopsies of healthy Fallopian tube, ovarian cancer (OvC) and omentum metastasis (OmM) cases comprising the Cambridge ovarian cancer TMA. There were no significant increases (ns) in LGR5 expression between the Fallopian tube and OvC and OmM samples. All 69 biopsies were scored in a single experiment presented as mean expression levels, +/− SEM. (F) Quantitation of LGR5 expression levels in samples from the Cambridge Brain cancer TMA- healthy brain tissue (Brain), low-grade glioma (LGG) and glioblastoma (GBM). There were no significant increases (ns) in LGR5 expression between Brain, LGG or GBM biopsies. A total of 15 biopsies were scored in a single experiment and presented as mean expression, +/− SEM. (G) Representative example of relative LGR5 protein expression levels in CD4+ T cells, CD8+ T cells and CD19+ B cells from healthy donor PBMCs determined by flow cytometry. A small number of CD19+ cells, <3% of the total population, express low levels of LGR5. The data shown is representative of three independent experiments using samples from five different healthy donors. (H) *LGR5* transcript levels normalised to *TBP*, measured by quantitative RT-PCR, in healthy donor B cells (B cells; 10 samples), B-ALL cell lines (Cell lines; 9 lines), CD19-enriched cell populations from primary B-ALL cases (ALL 1°; 22 samples) and CD19-enriched populations from B-ALL tumour cells maintained as PDX models (ALL-PDX; 15 samples). Arrowhead, expression levels of the NALM6, REH and 697 cell lines as well as the CRH, LC2 and SQ1 patient samples. Data presented as mean expression, +/− SEM. (I) LGR5 protein expression in the CRH, LC2 and SQ1 patient samples determined by flow cytometry. Histograms show detection with either Fl-α-LGR5 (red), Fl-α-LGR5 pre-incubated with blocking Frag1A (light grey) or with isotype control (dark grey). The data shown represent two independent experiments for CRH and LC2 sample and one experiment from the SQ1 sample. Source data are available online for this figure.

note, we observe approximately fivefold higher levels of cortical β-catenin in cell clusters corresponding to bile ducts in both control liver and HCC samples (Appendix Fig. S1D). Interestingly, the cell clusters associated with HCC tumours consistently expressed 3–5-fold lower levels of LGR5 relative to adjacent cells (Appendix Fig. S1D).

We were able to distinguish HCC tumoral sub-groups based on LGR5 protein expression: the non-proliferation sub-class of HCC is dominated by activating mutations in the Wnt pathway component β-catenin (or in rare cases AXIN1), lack of AFP expression and relatively low proliferation rates (Zucman-rossi et al, 2015). Across the cohort, we find negative correlations between LGR5 expression and serum AFP at the time of transplant (Appendix Fig. S1F) and the tumour cell proliferative fraction by Ki67 staining (Appendix Fig. S1G). Therefore, LGR5 expression is elevated in tumours with clinical and molecular features of the non-proliferative HCC sub-class.

Our transcription data ranked ovarian cancer as one of the highest LGR5-expressing cancers. However, owing to lack of data for fallopian tubes in the TCGA database, the presumptive tissue of origin for ovarian cancer, we were unable to determine whether this represents malignancy-specific increases in LGR5 overexpression. We probed 24 fallopian tube biopsies with α-LGR5 and α–β-catenin alongside a TMA containing 28 ovarian cancer and 14 omentum metastasis cases (Appendix Fig. S1H). Overall, LGR5 levels are low with five of the ovarian cancer and four of the omentum metastasis cases displaying elevated levels of protein in greater than 20% of tumour cells (corresponding to expression score of 1 or greater) (Fig. 2E; Appendix Fig. S1H). While there was a slight overall increase in LGR5 protein levels in ovarian cancer relative to fallopian tube epithelia we did not observe the corresponding increase for the omentum metastasis cases (Fig. 2E; Appendix Fig. S1H). β-catenin is expressed at approximately equal levels in epithelial and ovarian/omentum cancers (Appendix Fig. S1I).

LGR5 transcript levels in healthy pancreas and pancreatic cancer were not significantly different (Appendix Fig. S1B). When assessed for LGR5 protein levels, five matched healthy and tumour samples show no significant difference and were generally low, with three out of five healthy tissues and cancers having an expression score of 0.5 or 1 and the remaining samples scored as undetected (Appendix Fig. S1J). We obtained similar results with LGR5 expression in brain tissue and brain cancers (glioblastoma and low-grade glioma; LGG). LGR5 protein expression was low but detectable in healthy brain samples but only in 1/5 of the LGG and 2/5 of the GBM cases (Fig. 2F; Appendix Fig. S1K). We detected no overall changes in expression between healthy and malignant brain tissues.

A previous study has established a functional role for LGR5 in murine B-cell development and correlated LGR5 overexpression with a poor clinical outcome in human pre-B-ALL (Cosgun et al, 2017, 2020). Thus, we extended our expression analyses to immune cells and leukaemia. Flow cytometric analysis of human peripheral blood mononuclear cells (PBMCs) using α-LGR5 showed no expression of LGR5 in CD4+ and CD8+ T cells; however, we identified a small proportion (<1%) of CD19+ B cells with potentially low levels of LGR5 expression (Fig. 2G). To determine whether LGR5 levels are increased in leukaemia, we quantified LGR5 expression in patient-derived, primary acute lymphoblastic leukaemia (ALL), ALL patient-derived xenograft (PDX) models and pre-B-ALL cell lines. In the patient-derived samples (ALL and PDX), we found that 32 out of the 35 cases (90%) showed a significant increase in LGR5 mRNA levels with two cases displaying greater than 100-fold increases compared with healthy B-cell controls (Fig. 2H). We also observe significantly elevated LGR5 transcript levels in two out of nine pre-B-ALL cell lines; for instance, NALM6 cells had ~10,000× greater expression than the mean of healthy B cells (Fig. 2H). We chose three of the B-ALL cases (CRH, LC2 and SQ1) for further analysis of LGR5 protein levels using flow cytometry. While we observed differences in LGR5 protein levels amongst the B-ALL cases, there was a direct correlation between protein and relative transcript levels of LGR5 in the order CRH > LC2 > SQ1 (compare Fig. 2I,H).

Taken together, our analyses of LGR5 transcript and protein levels in healthy tissues and malignancies establish that: (i) LGR5 overexpression is specific to a discrete set of cancer types; (ii) there

is a substantial window of LGR5 expression between CRC, HCC, some ALL cases and healthy tissue; and (iii) the use of specific and well-validated antibodies such as α-LGR5 are a valuable diagnostic tool.

## α-LGR5 specifically detects LGR5 protein in human cancer cell lines

We next determined cellular expression levels of LGR5 protein in three human pre-B-ALL cell lines by western blot -NALM6 cells expressed the highest LGR5 protein levels followed by REH cells, while 697 cells expressed the least LGR5 protein levels (Fig. 3A,B). We next determined endogenous localisation of LGR5 using immunofluorescence of NALM6 cells and, consistent with over-expressed LGR5 in HEK293T cells and tissue and tumour biopsies, LGR5 localised to intracellular puncta with low observable signal associated with the cell periphery (Fig. 3C). To determine whether there were detectable levels of LGR5 on the cell surface of live NALM6 cells at steady state, we carried out flow cytometric staining at 4 °C using α-LGR5 labelled with the Dylight650 fluorophore (Fl- α-LGR5). While only 5% of cells harboured LGR5 surface expression when incubation was carried out at 4 °C, labelling for 1-h at 37 °C led to the majority of NALM6 cells acquiring the Fl-α-LGR5 probe (Fig. 3D). Indeed, analogous experiments carried out for REH and 697 cells with Fl-α-LGR5 incubations for 1-h at 37 °C were consistent with our expression analysis with REH displaying lower levels of fluorescence than NALM6 cells and 697 cells lacking detectable fluorescence (Fig. 3E). Our data indicate that LGR5 is predominantly expressed in intracellular puncta and a small transient pool of plasma membrane-associated that is internalised.

The LoVo colon cancer cell line has previously been shown to express sufficient levels of LGR5 for antibody detection (Gong et al, 2016; Junttila et al, 2015). Indeed, amongst a panel of five additional colorectal cancer cell lines (HCT116, SW480, RKO, DLD1 and HT29), LoVo cells were amongst the higher LGR5 transcript and protein expressors (Fig. 3F,G). Immunofluorescent detection of LGR5 in LoVo cells using Fl-α-LGR5 indicated expression was mainly confined to intracellular puncta with little detectable LGR5 on at the cell surface (Fig. 3H). As with NALM6 cells, flow cytometric detection of LGR5 expression in live LoVo cells required pre-incubation of Fl-α-LGR5 for 60 min at 37 °C to enable LGR5 internalisation (Fig. 3I). In contrast, we did not observe Fl-α-LGR5 internalisation by SW480 cells, likely the consequence of lower LGR5 protein expression levels (Fig. 3I).

Quantification of LGR5 expression in HCC cell lines (Fig. 3J,K) showed the highest LGR5 transcript and protein levels for HepG2 cells. Hep3B cells have high LGR5 transcript levels and slightly less LGR5 protein levels while PLC/PRF5 cells had lowest LGR5 RNA but high LGR5 protein expression. Interestingly, the HepG2 cell line harbours an oncogenic β-catenin mutation and may potentially serve as a cellular model for the non-proliferative HCC sub-class (Appendix Fig. S1F).

Our cell expression data establishes pre-B-ALL, CRC and HCC cellular models for LGR5-expressing cancer cells. The characteristic punctate distribution of LGR5 we observe in LoVo and NALM6 cells is consistent with our observations in CRC and HCC tumours. This distribution is due to rapid internalisation of plasma membrane targeted LGR5 into intracellular puncta.

## Rapid internalisation of α-LGR5 antibodies by LGR5-overexpressing cell lines

To determine the kinetics of LGR5 internalisation we treated hLGR5-eGFP overexpressing HEK293T cells with Fl-α-LGR5: within 5 min Fl-α-LGR5 was internalised and associated with puncta juxtaposed to the cell surface. Ultimately, the internalised signal from Fl-α-LGR5 associated entirely with the intracellular hLGR5-eGFP puncta over the 45-min experimental time course (Fig. 4A). By contrast, HEK293T cells transfected with LGR4-eGFP failed to internalise FL-α-LGR5 (Fig. 4A).

Interestingly, the rate of endogenous LGR5 internalisation by NALM6 and LoVo cells was identical to internalisation by hLGR5-eGFP-overexpressing HEK293T cells: Fl-α-LGR5-associated intra-cellular puncta in both NALM6 and LoVo cells were evident within 5 min and increased throughout the 60-min time course for both NALM6 (Fig. 4B) and LoVo cells (Appendix Fig. S2A).

We next compared α-LGR5 antibody internalisation in cells expressing endogenous levels of LGR5 relative to internalisation of the HER2 cell surface receptor using the well-validated, commercial Trastuzumab antibody (α-HER2). For these and subsequent studies, we used the humanised version of α-LGR5 (α-LGR5v4), derived from the murine α-LGR5 that retained high-affinity binding to the LGR5 epitope, $Kd = 2$ nM, on par with the affinity of the parental murine antibody (Table 1). Likewise, α-LGR5v4 detects HEK293T cells overexpressing human and *cynomolgus* LGR5 proteins but not the murine LGR5 (Appendix Fig. S2B). As a negative control, we used a non-binding version of the antibody generated during the humanisation process (α-LGR5v6) which showed no detectable LGR5 binding (Table 1; Appendix Fig. S2B).

We created fluorescently labelled versions of α-LGR5v4 (Fl-α-LGR5v4) and a version of Trastuzumab conjugated to a different fluorophore (Fl-α-HER2) for simultaneous detection of the two receptors. Incubation of LoVo cells with Fl-α-LGR5v4 and Fl-α-HER2 for 60 min at 37 °C led to internalisation of both fluorophores, detected by flow cytometry; however, NALM6 cells are only able to internalise Fl-α-LGR5v4 and not Fl-α-HER2 (Fig. 4C; Appendix Fig. S3A) indicating a lack of HER2 expression by this cell line. A fluorescent version of the control Fl-α-LGR5v6 and the corresponding isotype control showed significantly less binding to LoVo and NALM6 cells (Appendix Fig. S3A). Moreover, internalisation of Fl-α-LGR5v4 by either LoVo or NALM6 cells was abrogated by pre-incubation of the antibody with Frag1A (Appendix Fig. S3B).

The co-expressed LGR5 and HER2 in LoVo cells enabled us to directly compare the kinetics of endogenous LGR5 internalisation relative to the HER2 receptor by incubating LoVo cells with Fl-α-LGR5v4 and Fl-α-HER2 over a 180-min time course. We used unbiased automated scoring in a three-dimensional field of view for classifying total Fl-α-LGR5v4 and Fl-α-HER2 bound to the surface, 'associated', or 'internalised' by LoVo cells, by the appearance of intracellular Fl-α-LGR5v4 and Fl-α-HER2 puncta within the cortical actin cell periphery. At 5 and 15 min, while Fl-α-HER2 associated with 70% of all LoVo cells, Fl-α-LGR5v4 associated with significantly more LoVo cells (>90%) (Fig. 4D; Appendix Fig. S3C). Importantly, the internalisation rates for Fl-α-LGR5v4 and Fl-α-HER2 were also distinct throughout the time course—100% of the cells associated with Fl-α-LGR5v4 signal showed internalisation within 5 min whereas only 13% of cells internalised Fl-α-HER2

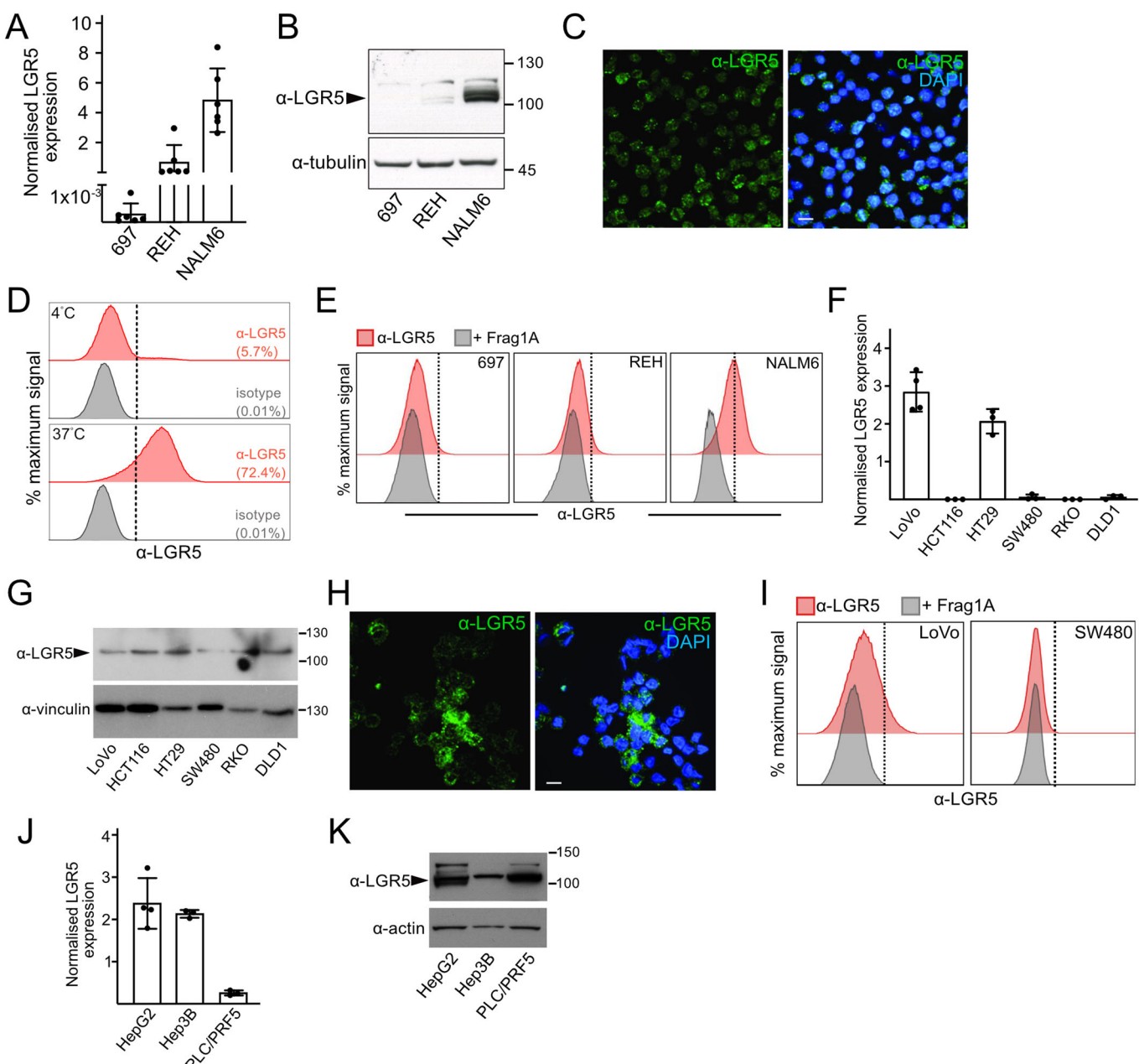

**Figure 3. Characterisation of LGR5 expression in pre-B-ALL and CRC cell lines.**

(A) Relative *LGR5* transcript levels in pre-B-ALL cell lines measured by quantitative RT-PCR with *TBP* as a reference gene. Data is presented as mean expression, error bars represent +/− standard deviation (SD) for six biological replicates. (B) Western blot analysis of LGR5 protein levels in lysates from pre-B-ALL cell lines using an antibody to tubulin as loading control. Data represents two independent experiments. (C) Indirect immunofluorescence of NALM6 cells using α-LGR5. Scale bar, 5 μm. Data represents two independent experiments. (D) Flow cytometric detection of Fl- α-LGR5 (red histograms) or fluorescent isotype control (grey histograms) association with NALM6 cells after 60 min incubation at 4 °C (top panel) or 37 °C (bottom panel). Data represents 1 experiment conducted at 4 °C and 4 independent experiments conducted at 37 °C. (E) Flow cytometric analysis of 697, REH or NALM6 cells after 60 min incubation at 37 °C with Fl- α-LGR5 (red histograms) or Fl- α-LGR5 together with Frag1A (control, grey histograms). Data are representative of three independent experiments. (F) Relative *LGR5* transcript levels normalised to *TBP* in CRC cell lines measured by qRT-PCR. Data is presented as mean expression, +/− SD for three biological replicates except for LoVo cells, four biological replicates. Data are representative of 3–4 independent experiments. (G) Western blot analysis of LGR5 protein in CRC cell lines using α-LGR5 and an antibody to vinculin as loading control. The western blot is representative of three independent experiments. (H) Indirect immunofluorescence of LoVo cells using α-LGR5. Scale bar, 5 μm. Images are representative of two independent experiments. (I) Flow cytometric detection of Fl- α-LGR5 (red histograms) or Fl- α-LGR5 pre-incubated with Frag1A (grey histograms) association with LoVo and SW480 cells after 60 min incubation at 37 °C. Data represents a single experiment conducted using SW480 cells and four independent experiments with LoVo cells. (J) Relative *LGR5* transcript levels normalised to *TBP* in HCC cell lines. Data is presented as mean expression, +/− SD for 3–4 biological replicates. (K) Western blot analysis of LGR5 protein levels in HCC cell lines using an antibody to actin as a loading control. Western blots are representative of two independent experiments. Source data are available online for this figure.

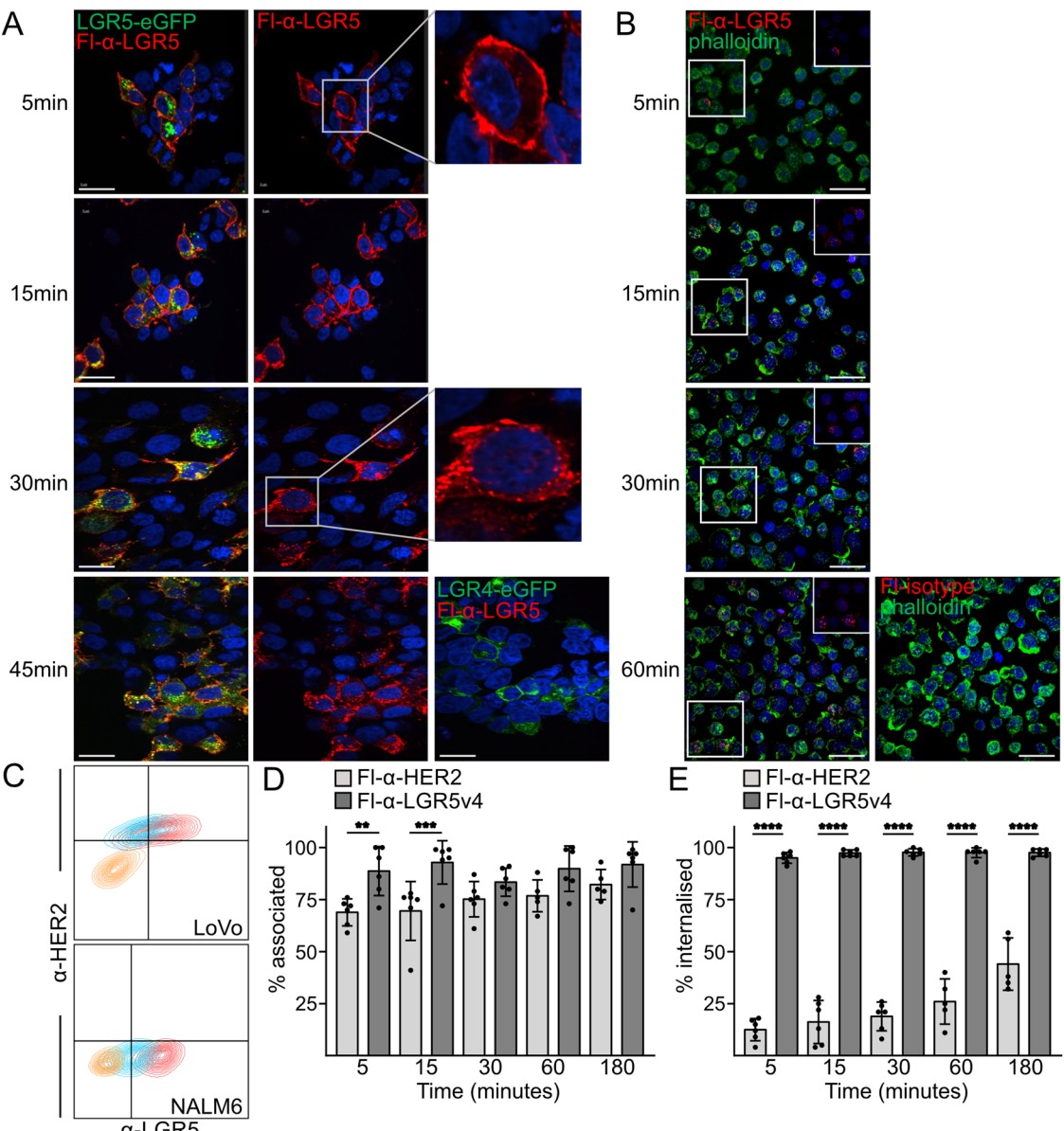

**Figure 4. Rapid internalisation of cell surface LGR5.**

(A) Time course of FL-α-LGR5 internalisation by LGR5-eGFP expressing HEK293T cells. Top right panel: enlarged image showing association of FL-α-LGR5 with the cell periphery after 5 min, followed by co-localisation with the internal LGR5-eGFP associated puncta within 30 min (third right panel). Bottom right panel: no association or internalisation of FL-α-LGR5 for cells expressing LGR4-eGFP. Internalisation data is representative of two independent experiments. Scale bar, 10 μm. (B) Time course of red fluorescent-labelled Fl-α-LGR5v4 (red) internalisation by NALM6 cells. Incubations were fixed at various timepoints and probed with fluorescent phalloidin (green) and Hoechst (blue). Insets (top right) show Fl-α-LGR5v4 (red) channel only. In the bottom right panel, a red fluorescent-labelled version of the control Fl-α-LGR5v6 was used and no internalisation was observed. Data are derived from a single experiment. Scale bar, 10 μm. (C) Flow cytometric analysis of LoVo and NALM6 cells after 60 min of incubation with Fl-IgG1 (orange), and Fl-α-HER2 (blue) with either Fl-α-LGR5v4 (red). Data represent a single experiment. Each panel is a composite of the individual analyses in Appendix Fig. S3B. (D) Time course of Fl-α-LGR5v4 and Fl-α-HER2 association with LoVo cells. Percent association was scored as fraction of cells associated with fluorescent signals from Fl-α-HER2 (light grey bars) or Fl-α-LGR5v4 (dark grey bars) at the indicated timepoints per field of view (example images shown in Appendix Fig. S3D). Datapoints are average of 6 separate scoring set ups, each measuring at 80–200 cells per condition over 2 independent experiments. Data is presented as mean expression, +/− SD amongst scoring experiments. (E) Time course of percent internalisation of Fl-α-LGR5v4 and Fl-α-HER2 by LoVo cells. Internalisation data scores 6 experimental counts of 80–200 cells over two independent experiments. Data are presented as mean expression, +/− SD. Source data are available online for this figure.

after 5 min, rising to 45% after 180 min (Fig. 4E; Appendix Fig. S3C).

We next traced the intracellular trafficking of internalised LGR5. Previous studies with overexpressed hLGR5-eGFP have established constitutive internalisation and trafficking to either a LAMP1-positive compartment (Gong et al, 2016) or the Golgi (Snyder et al, 2013). Internalisation of LGR5 is clathrin- and dynamin-dependent (Snyder et al, 2017) and is in part regulated by intracellular factors interacting with the C-terminus (Snyder et al, 2013). One study has shown interaction of LGR5 with the cytoskeletal regulator IQGAP1

that in turn influences cortical actin dynamics and cell-cell adhesion (Carmon et al, 2017). However, these cellular studies have relied on LGR5 transgene overexpression that may influence internalisation kinetics and cellular distribution. Thus, we pulsed LoVo and NALM6 cells with Fl-α-LGR5v4 for 60 min, fixed and probed the signal from the internalised antibody in relation to molecular probes of specific sub-cellular compartments (Appendix Fig. S3D). Analysis of co-localisation data from these studies (representative examples of imaging data analysed for LoVo and NALM6 cells are found in Appendix Fig. S3E,F, respectively). For both cell types, Fl-α-LGR5v4 was associated with all markers of intracellular compartments tested, in particular with the endosomal recycling markers SNX1 and SNX27 for LoVo cells (Appendix Fig. S3G) and the constitutive internalisation marker CD71 for NALM6 cells (Appendix Fig. S3H). Importantly, Fl-α-LGR5v4 is targeted to the LAMP1-positive lysosomal compartment in both LoVo and NALM6 cells (Appendix Fig. S3G,H).

Taken together, Fl-α-LGR5v4 internalisation and distribution studies indicate that endogenously expressed LGR5 is rapidly internalised from the plasma membrane by the endocytic pathway and is both recycled to the plasma membrane and directed to lysosomal vesicles.

## Validation of α-LGR5-based antibody-drug conjugates

The high LGR5 expression levels specific to certain malignancies, rapid internalisation kinetics of α-LGR5 and LGR5 trafficking to the lysosome raised the intriguing prospect of targeting cancers using α-LGR5-based antibody-drug conjugates (ADCs). Indeed, two previous studies have used ADC versions of two other LGR5 antibodies for killing LoVo cells in vitro and targeting the LoVo cell tumour model in vivo (Junttila et al, 2015; Gong et al, 2016). We conjugated the microtubule poison MMAE to the α-LGR5 murine clone, α-LGR5v4 and α-LGR5v6. For α-LGR5, we generated both the sulphatase cleavable version (α-LGR5-ADC; (Bargh et al, 2020; Walsh et al, 2019)) and the non-cleavable version (α-LGR5-ADC$^{NC}$) (Walsh et al, 2019). We also generated the IgG1-MMAE conjugate in the cleavable linker format (IgG1-ADC) as a control for our cell-killing studies. α-LGR5-ADC and α-LGR5v4-ADC demonstrate near identical epitope binding affinities to the parental antibodies (Table 1).

Treatment of NALM6 cells with α-LGR5-ADC for 3 days lead to effective cell killing, with an EC50 of 4 nM (Fig. 5A). α-LGR5-ADC is slightly less effective against REH pre-B-ALL cells that express lower LGR5 levels (Fig. 3A,B) with an EC50 of 10 nM. We find no effect on cell viability when treating NALM6 cells with α-LGR5-ADC$^{NC}$ consistent with a previous study that found a non-cleavable version of an LGR5 antibody-based ADC was ineffective at cell killing (Gong et al, 2016).

We also test the ability of α-LGR5-ADC to kill LoVo cells, again using α-LGR5-ADC$^{NC}$ as a specificity control. During the three-day incubation, α-LGR5-ADC effectively kills LoVo cells with an EC50 value of 9 nM (Fig. 5B). In contrast, treatment with α-LGR5-ADC$^{NC}$ at concentrations up to 50 nM does not lead to cell death. We conclude that α-LGR5-ADC is highly specific and effectively kills NALM6 and LoVo cells owing to high levels of LGR5 expression.

We next tested the ability of the α-LGR5v4-ADC to target 4 CRC organoid models expressing variable LGR5 expression levels. The four individual models express relatively equivalent LGR5

protein and transcript levels (Fig. EV2A,B) that increase in expression from CRC1 to CRC4. Intriguingly, we observe increasing sensitivity of the four organoid models to α-LGR5v4-ADC treatment (Fig. EV2C) that matches the relative increase in LGR5 expression levels.

## α-LGR5-ADC targets NALM6 tumours in vivo

We next determined the therapeutic utility of α-LGR5 in vivo by testing the ability of α-LGR5-ADC to target NALM6 cells stably expressing luciferase in NSG mice. On day 5 post implantation (PI), IVIS imaging was used to classify tumour-bearing mice into two groups with equal overall tumour burden. On days 6, 8, 10 and 12 PI, mice were treated intravenously (IV) with 5 mg/kg α-LGR5-ADC via tail vein injection (Fig. 5C). The control cohort of mice received injections of 5 mg/kg IgG1-ADC control on these days. Tumour burden was monitored at 2–3-day intervals by IVIS imaging. We find that while NALM6 tumours treated with IgG1-ADC grow at a logarithmic rate, α-LGR5-ADC treatment leads to rapid tumour regression within 4 days of the initial injection of α-LGR5-ADC (Fig. 5C). Tumour regression persists throughout the course of the four treatments after which tumour growth resumes with a 4-day latency period. IVIS imaging on day 19 indicates that the α-LGR5-ADC treatment group maintains less than 0.5% of the tumour burden of control mice (Fig. 5C,D). At experimental endpoint on day 20, we also note a marked reduction in splenic mass (approximately twofold) and residual splenic NALM6 cells (approximately 100-fold) as well as a reduction in absolute numbers of NALM6 cells in the blood (100-fold) and bone marrow (50-fold) of α-LGR5-ADC-treated mice (Figs. 5E). We did not observe adverse effects of ADC treatment on the distribution of ß-catenin distribution or crypt proliferation in sections of small intestinal epithelia relative to untreated mice (Appendix Fig. S4A,B).

To determine whether the humanised α-LGR5 antibody could also be used therapeutically, we treated NALM6 tumour-bearing mice with two doses of 5 mg/kg α-LGR5v4-ADC or 5 mg/kg of the non-binding control α-LGR5v6-ADC on days 6 and 8 post tumour implantation (PI). Consistent with our previous in vivo trial, tumour growth is largely arrested after a 4-day latency period from the first treatment day. The tumour growth arrest persists for at least 4 days after the last α-LGR5v4 treatment after which point tumour growth resumes (Fig. 5F). We find that overall NALM6 tumour burden assessed by IVIS imaging on day 19 for the α-LGR5v4-ADC-treated mice is about half of the control values (Fig. 5F,G). As seen with murine α-LGR5-ADC, we note at experimental endpoint on day 20 a reduction in splenic mass (threefold) and associated residual splenic NALM6 cell numbers (~50-fold) as well as an approximately tenfold reduction of NALM6 cells in the blood (Fig. 5H). Interestingly, this more modest treatment regime reveals that NALM6 cells were fully retained in the bone marrow where there were no significant differences in total tumour cell number between α-LGR5-ADC treated and control-treated animals (Fig. 5H). Altogether, our in vivo trial data indicates that the α-LGR5-ADCs are effective at diminishing tumour growth, but persistent treatment is required for a durable response.

## α-LGR5-based BiTE molecules are functional and show efficacy in vivo

To test the potential for creating other α-LGR5-based therapeutic modalities, we generated the humanised α-LGR5 scFv fragment

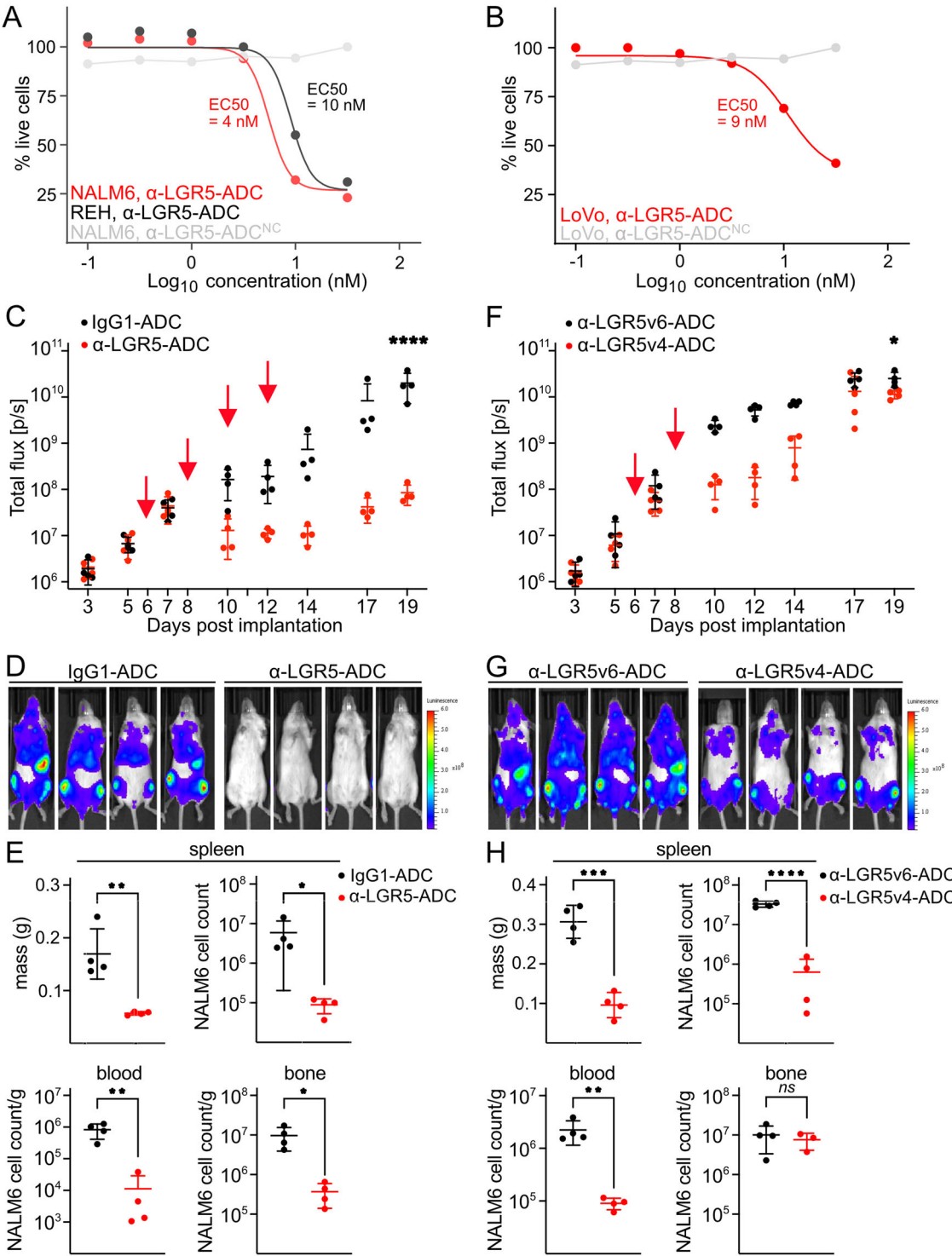

(α-LGR5scFv) version of α-LGR5. The His-tagged α-LGR5scFv purified from expressing HEK293F cells (Appendix Fig. S5A) retained high-affinity binding to its LGR5 epitope, with a *Kd* value of 0.77 nM (Table 1). α-LGR5scFv was used to create FLAG-tagged BiTE constructs in two formats—the N-terminal α-CD3scFv fused to α-LGR5scFv (CL-BiTE) and the N-terminal α-LGR5scFv fused to α-CD3scFv (LC-BiTE), produced from expression in HEK293F cells and purification via α-FLAG affinity chromatography (Appendix

Fig. S5A). First, we asked whether the BiTE molecules could efficiently activate human T cells using healthy donor PBMCs as a source of CD4+ or CD8+ T cells. T-cell activation was measured by combined expression of the T-cell activation markers CD25 and CD69 in the absence and presence of target NALM6 cells. We observed minimal levels of CD4+ T and CD8+ T-cell activation in the presence of LC-BiTE and CL-BiTE alone, but potent T-cell activation with the addition of target NALM6 cells: approximately

**Figure 5.   Targeting of cancer cells with α-LGR5-ADC.**

(A) Survival of NALM6 or REH cell lines after 72 h treatment with α-LGR5-ADC. Survival data was fit to a non-linear EC50 shift model yielding EC50 values of 4 and 10 nM, respectively. As control, NALM6 cells were treated with the non-cleavable α-LGR5-ADC[NC] which did not reduce the cell count after 72 h. Data represent two independent experiments for each treatment. (B) LoVo cells were treated with either α-LGR5-ADC or the non-cleavable α-LGR5-ADC[NC]. Modelling of cell survival data to a non-linear EC50 shift model yielded an EC50 of 9 nM. Data represent two independent experiments for each treatment. (C) Quantification of NALM6 tumour size measured by IVIS imaging over the course of the treatment with either 5 mg/kg α-LGR5-ADC or IgG1-ADC control. Red arrows indicate treatments on days 6, 8, 10 and 12 PI. Data are presented as mean expression, +/− SD at each timepoint. Data represents $n = 4$ mice for each treatment group. (D) IVIS images of IgG1-ADC and α-LGR5-ADC-treated mice, ventral view, on day 19 post implantation. (E) Top left: spleen mass and absolute number of NALM6 cells extracted from spleen (top right) of treated mice at experimental endpoint, day 20 PI. Bottom left: density of NAML6 cells in blood circulation; bottom right—NALM6 cells recovered from bone marrow. Data are presented as mean expression, +/− SD for data from $n = 4$ mice per treatment group. (F) Quantification of NALM6 tumour size by IVIS imaging over the course of the treatment with either 5 mg/Kg α-LGR5v4-ADC or α-LGR5v6-ADC. Antibody symbols indicate treatments on days 6 and 8 post implantation. Data are presented as mean expression, +/− SD at each timepoint for data from $n = 4$ mice for each treatment group. (G) IVIS images of α-LGR5v4-ADC or α-LGR5v6-ADC-treated mice, ventral view, at the experimental endpoint. (H) Top left: spleen mass and absolute number of NALM6 cells extracted from spleen (top right) of treated mice at experimental endpoint. Bottom left: density of NALM6 cells in blood circulation; bottom right—NALM6 cells recovered from bone marrow. Data is presented as mean expression, +/− SD for data from $n = 4$ mice per treatment group. ns no significant difference between treated cohorts. Source data are available online for this figure.

4-fold increase in CD4[+] T-cell activation with LC-BiTE or CL-BiTE addition, and a fivefold (LC-BiTE) or 12-fold (CL-BiTE) increase in CD8[+] T-cell activation over the course of 24 h (Fig. 6A,B). Importantly, we observed effective and specific tumour cell killing when NALM6 cells and cytotoxic CD8[+] T cells were incubated with LC-BiTE (20%) or CL-BiTE (45%), respectively, during a 6-h time course (Fig. 6C). Similarly, we found a potent, differential CL-BiTE-induced killing of high LGR5 expressing NALM6 preB-ALL and LoVo CRC cell lines versus their low LGR5 expressing counterparts (697 and SW480 cell lines) (EV3A,B), as well as between the patient derived pre-ALL models CRH (high LGR5 expressing) and LC2 (low LGR5 expressing) (EV3C).

We next tested the performance of CL-BiTE in vivo which necessitated an alternative method for scaling up protein production. We generated highly pure CL-BiTE from bacterial expression that was effective in targeting NALM6 cells in vitro, with a similar activity to the CL-BiTE produced from HEK293F expression (Appendix Fig. S5B). CL-BiTE was used to target the in vivo NALM6 tumour model; two doses of 100 µg in conjunction with $7–10 × 10^6$ PBMCs were used to treat NALM6 tumour-bearing mice on day 3 and 7 post tumour implantation. On day 11 post implantation, we observed a small but significant decrease in collective tumour burden in CL-BiTE treated mice, approximately twofold (Fig. 6D,E). The high residual tumour burden left in the CL-BiTE-treated experimental group enabled us to assess LGR5 expression levels in the tumour cells. We imaged consecutive tissue sections of residual tumour cells in spleens of CL-BiTE treated animals by immunohistochemistry (IHC) using α-LGR5 and an antibody to human CD20 to mark NALM6 tumour cells. We found that tumour cells in CL-BiTE-treated animals retain expression of LGR5 protein to the same extent as control-treated animals (Fig. 6F). We conclude that residual tumour load from treatment of tumour-bearing mice with α-LGR5-based therapeutics are the consequence of incomplete tumour access and targeting and not due to downregulation of LGR5 expression in tumour cells. Taken together, our data support further development of the CL-BiTE modality.

## α-LGR5-based cell therapies effectively target NALM6 cells

The α-LGR5[scFv] was used for the generation of a second generation CAR construct that contained the CD28 co-stimulatory module (LGR5[scFv]-CAR) (Zhong et al, 2010). Lentiviruses encoding the

CAR construct were used to infect CD8[+] T cells derived from human PBMCs as well as the human Natural Killer cell line NK92 that has recently shown potential as an 'off-the-shelf' CAR vehicle for therapy (Siegler et al, 2018; Mitwasi et al, 2020).

We tested the ability of LGR5[scFv]-CAR-T cells to target HEK293T cells expressing LGR family transgenes. LGR5[scFv]-CAR-T cells specifically killed HEK293T cells expressing the hLGR5-eGFP and *cynomolgus* LGR5-eGFP transgene after 9 h of co-culture, which was approximately eightfold greater than the baseline levels of killing activity observed with CD8[+] T-cell incubation. The killing activity of LGR5[scFv]-CAR-T cells towards HEK293T cells expressing the *cynomolgus* LGR5-eGFP transgene was at approximately 85% of the killing potency against hLGR5-eGFP-expressing HEK293T cells likely due to the 2aa changes in the binding site (Fig. 7A). No specific killing by LGR5[scFv]-CAR-T cells was observed in the presence of HEK293T cells expressing murine Lgr5 or human LGR4/6 transgenes (Fig. 7A). Similarly, LGR5[scFv]-CAR-NK92 cells showed exquisite specificity towards HEK293T expressing the hLGR5-eGFP and *cynomolgus* LGR5-eGFP transgene and no specific activity against HEK293T cells expressing mLgr5, hLGR4 or hLGR6 transgenes (Appendix Fig. S6A).

Next, LGR5[scFv]-CAR-T-cell cytotoxicity was measured over time for targeting the NALM6, HepG2 and LoVo cell lines that express high endogenous levels of LGR5. We observed rapid tumour cell destruction by LGR5[scFv]-CAR-T cells within 6 h of incubation with NALM6 and LoVo cells and HepG2 cells. Killing assays conducted with CD8[+] T-cell effector cells alone elicited some tumour cell death particularly in the HepG2 line. (Fig. 7B). However, the LGR5[scFv]-CAR-T cells showed approximately threefold greater specific killing activity. We also observed enhanced differential killing of high LGR5 versus low LGR5 expressing preB-ALL cell lines (NALM6 and 697 cells, respectively; EV4A) and CRC cell lines (LoVo and SW480 cells, respectively; EV4B) by LGR5[scFv]-CAR-T cells. Similarly, the high LGR5 expressing CRH patient-derived preB-ALL model demonstrated enhanced sensitivity to LGR5[scFv]-CAR-T cell killing versus T cells alone, whereas the low LGR5 expressing LC2 model was equally sensitive to LGR5[scFv]-CAR-T cell and T cell killing (EV4C). LGR5[scFv]-CAR-NK92 cells also showed effective targeting of NALM6 cells with an approximately twofold increase over killing by NK92 cells over the course of 9 h (Appendix Fig. S6B). Interestingly, in the same time window REH cells, a preB-ALL line with low expression of LGR5, was not targeted by LGR5[scFv]-CAR-NK92 cells indicating that a safe therapeutic window could be achieved for the LGR5[scFv]-CAR (Appendix Fig. S6B).

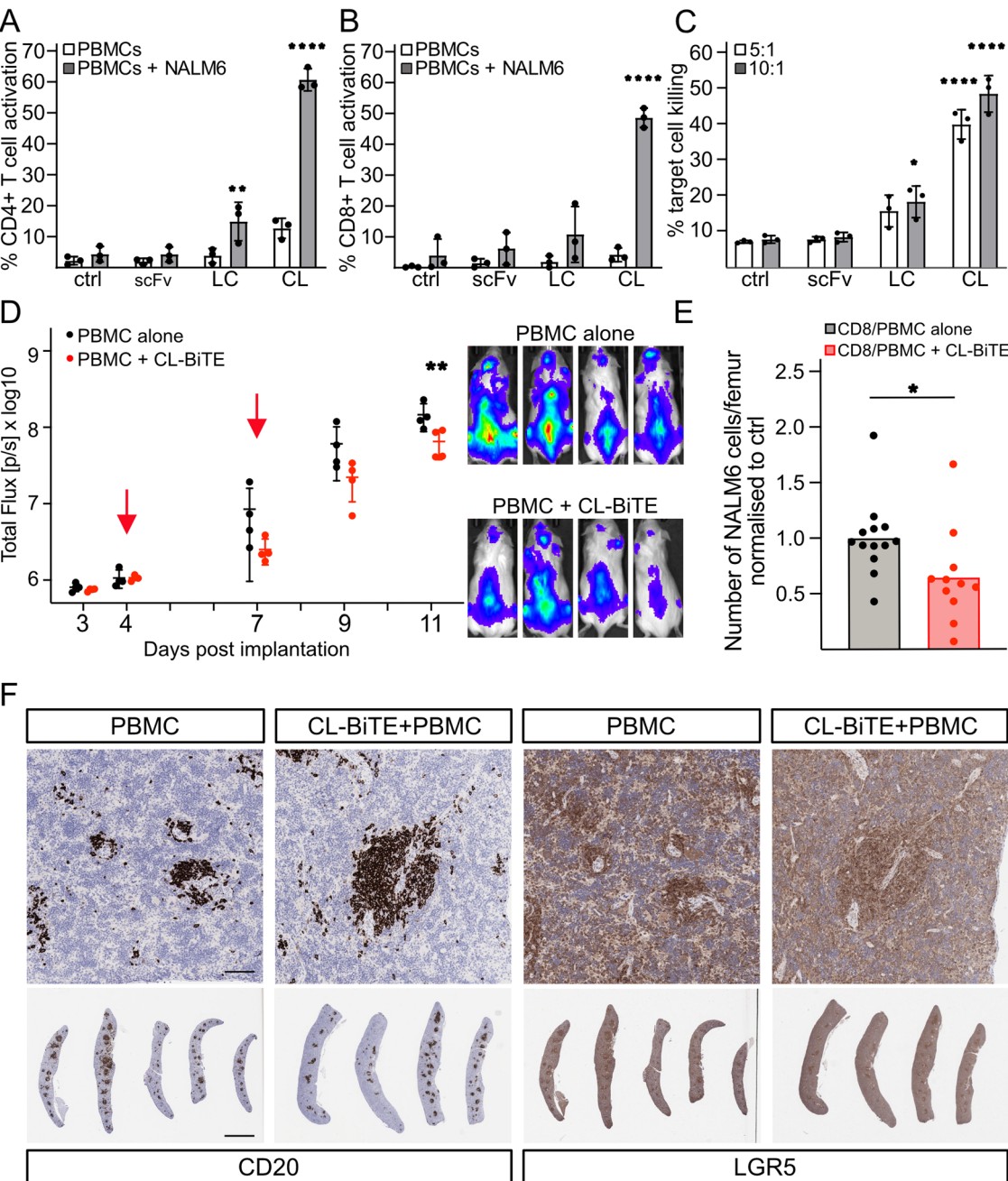

**Figure 6. In vitro killing activity and in vivo efficacy performance of α-LGR5-based BiTE molecules.**

(A) CL- and LC-mediated activation of CD4$^+$ T cells in healthy donor PBMCs in the presence or absence of NALM6 target cells determined as percent cells with combined expression of CD25 and CD69. Control (ctrl), no addition of molecule, scFv refers to treatment with the α-LGR5$^{scFv}$ fragment. Data are presented as mean expression, $+/-$ SD from two independent experiments using three different healthy donor PBMCs. (B) CL- and LC-mediated activation of CD8$^+$ T cells in healthy donor PBMCs in the presence or absence of NALM6 target cells determined as percent cells with combined expression of CD25 and CD69. scFv is the control α-LGR5$^{scFv}$ fragment. Data is presented as mean expression, $+/-$ SD of two independent experiments using three different healthy donor PBMCs. (C) NALM6 target cell killing by cytotoxic CD8$^+$ T cells in the absence (ctrl) or with the addition of scFv control or LC and CL-BiTEs, respectively. Killing was assessed after 6 h at effector to target cell ratios of 5:1 and 10:1. Data shown are from $n = 4$ independent experiments using three donors and is presented as mean expression, $+/-$ SD. (D) In vivo efficacy of CL-BiTE was determined in the NALM6 tumour model. Mice were treated with two doses of 100 μg CL-BiTE each and 7–10 × 10$^6$ PBMCs at days 3 and 7 post implantation. Tumour size was measured by IVIS imaging over the course of the treatment. Control mice were treated with 7–10 × 10$^6$ PBMCs in the absence of CL-BiTE. Data are presented as mean expression, $+/-$ SD. Right panels: IVIS images of control and treated mice on day 11 post implantation. Data for 2 independent experiments is shown. (E) Numbers of NALM6 cells were determined per femur at endpoint on day 11 post implantation. Data shown are pooled from two independent experiments: one set treated with CD8 T cells $+/-$ CL-BiTE and the other with PBMCs $+/-$ CL-BiTE and normalised to the respective PBMC or CD8$^+$ T-cell alone control. Data are presented as mean expression, $+/-$ SD for $n = 11$–13 mice per condition. (F) Assessment of LGR5 protein levels in splenic NALM6 tumour cells post CL-BiTE treatment on day 11. Consecutive sections of spleens from mice treated with PBMCs or PBMCs plus CL-BiTE were imaged using an antibody to the human B-cell marker CD20 (to discriminate NALM6 cells) and α-LGR5. Data is presented as mean expression, $+/-$ SD and is representative of two independent experiments. Scale bar: 2 mm (overview). Scale bar: 100 μm (zoom). Source data are available online for this figure.

We next carried out in vivo efficacy trials for LGR5$^{scFv}$-CAR-T cells following a similar protocol to the in vivo α-LGR5-ADC and BiTE experiments. We injected $3–5 \times 10^6$ LGR5$^{scFv}$-CAR-T cells at days 4 and 7 PI alongside T-cell and no T-cell (PBS) control groups with tumour development monitored by IVIS imaging until experimental endpoint on day 11 (Fig. 7C). We observed little to no difference in tumour sizes after the first treatment with LGR5$^{scFv}$-CAR-T cells, but after the second injection of LGR5$^{scFv}$-CAR-T cells PI we observed an approximate fivefold decrease on day 10 in NALM6 tumour size for the LGR5$^{scFv}$-CAR-T-cell treatment group relative to the T-cell and no T-cell control groups, and approximately threefold on day 11 (Fig. 7C,D). Throughout the trial, there were no significant differences in tumour burden between the T-cell and no T-cell control groups. Enumeration of residual NALM6 cells at the experimental endpoint indicated significant depletion in tumour cell numbers in the bone marrow in animals treated with the LGR5$^{scFv}$-CAR-T cell (Fig. 7E). The high specificity and efficacy demonstrated by the LGR5$^{scFv}$-CAR showcases the potential for effective targeting of a range of cancer types.

## Discussion

LGR5 is an established stem cell marker in a number of murine tissues (Leung et al, 2018; Haegebarth and Clevers, 2009; Nusse and Clevers, 2017), is overexpressed in a range of cancers (Morgan et al, 2018; Yamamoto et al, 2003; McClanahan et al, 2006; Tanese et al, 2008; Hagerling et al, 2020; Nakata et al, 2013; Cosgun et al, 2017, 2020), and studies using genetically engineered mouse models (GEMMs) and human CRC organoids have found that LGR5 is critical for maintaining the proliferative compartment in tumours (Junttila et al, 2015; Gong et al, 2016; Morgan et al, 2018). It has been difficult however, to fully assess LGR5 as a therapeutic target for the treatment of cancer due to the scarcity of specific, high-affinity antibodies against human LGR5.

Here we report the development, validation and characterisation of novel antibodies raised against the extracellular domain of human LGR5 with a clear line of sight to therapeutic application. α-LGR5 is unique from previously reported α-LGR5 antibodies (Appendix Table S1) (Gong et al, 2016; Junttila et al, 2015; Herpers et al, 2020; De Lau et al, 2011) in the sequences of its CDR3 regions. We have fine-mapped the interaction for α-LGR5 on the LGR5 protein that resides within the N-terminal 15 amino acids of the extracellular domain, a disordered region of the protein that precedes the first LRR (Appendix Fig. S7). The epitope appears to be unique amongst the reported antibodies whose binding to LGR5 has been mapped to sites within LRRs 1–9 (Appendix Fig. S7) (Gong et al, 2016; Junttila et al, 2015; Herpers et al, 2020; De Lau et al, 2011). It is interesting to note that binding of α-LGR5 or any of the reported antibodies to LGR5 does not interfere with its function in potentiating Wnt pathway activity indicating that none of the target epitopes overlap with the R-spondin binding site. Structurally, α-LGR5 and MCLA-158 bind to a region at the N-terminus of the LGR5 extracellular domain distinct from the R-spondin binding site whereas 8F2, BNC101, CNA3103 and he8E11v2 are reported to bind to the convex portion of the extracellular domain (Appendix Fig. S7).

Another distinguishing feature of α-LGR5 is its high affinity to its epitope on LGR5—we carried out in vitro binding studies on the parental murine α-LGR5 antibodies, its humanised and ADC versions and the scFv fragments to enable direct comparison of epitope affinity values with the other reported antibodies—the α-LGR5 antibodies rank amongst the highest affinity binders of the reported antibodies (Appendix Table S1).

We have demonstrated that α-LGR5 is a highly versatile research tool compatible with a range of techniques such as western blot, immunofluorescence and flow cytometry and is a sensitive and specific reagent for determining cellular levels and localisation of LGR5 protein in healthy and malignant tissue sections. Importantly, we were able to conduct extended analysis of LGR5 protein expression, a critical extension to previous transcriptomic analyses (Junttila et al, 2015; Gong et al, 2016) in normal and malignant tissues. We note the prominent differences between the results obtained from ours and previous transcript analyses of LGR5 and α-LGR5-defined LGR5 protein expression exemplified in brain and ovarian cancers; both malignancies consistently score high for LGR5 transcript levels, however, we detected only low LGR5 protein levels in few cells, in less than 10% of the tumours. Moreover, LGR5 transcript levels in liver tissue and HCC are similar, yet LGR5 protein levels score much higher in malignancy. These disparities may be due to the ability to distinguish LGR5 expression in epithelial tissue by immune fluorescence versus bulk transcript levels in biopsies, the use of different tissue collection protocols amongst sample cohorts and poor correlation of RNA and protein. In line with this notion, we observed an excellent correlation between LGR5 transcript and protein levels for the 3 B-ALL-PDX models we tested, likely due to the fact we freshly isolated the B-ALL cells. Together, our data highlight the need for directly determining LGR5 protein levels in healthy and tumour tissues as a faithful indicator of tissue and tumour expression.

We have determined that elevated LGR5 expression is a defining characteristic of CRC, HCC and pre-B-ALL. Importantly, our tissue census indicates that healthy tissues harbour very low to undetectable LGR5 protein, paving the way for therapeutic strategies targeting malignancies that overexpress the protein. While we have established CRC, HCC and pre-B-ALL as priority cancer targets for α-LGR5-based therapeutics, future studies will determine other targetable cancer types by increased LGR5 protein levels as a prognostic marker. This has proven particularly relevant to the assessment of LGR5 protein levels in HCC. We found that high LGR5 protein expression further delineates the HCC subset with activating mutations in β-catenin, that are characterised by low T-cell infiltration and thus referred to as immune deserts (Berraondo et al, 2019; Galarreta et al, 2019). The prediction is that this HCC subset will be refractory to both checkpoint inhibition and cellular therapies. Indeed, reporting from the CheckMate 459 trial (NCT02576509) evaluating nivolumab (PD1 checkpoint inhibitor) versus sorafenib (small molecule kinase inhibitor) in HCC failed to meet its endpoint target of improved overall survival (Yau et al, 2022; Berraondo et al, 2019). However, we propose that improved patient stratification by exclusion of the high LGR5-expressing cohort may be important to reach the expected outcome of the trial. Taken together, the identification of HCC patients with high levels of LGR5 using α-LGR5 presents an intriguing biomarker opportunity that both reports the β-catenin mutant subset and

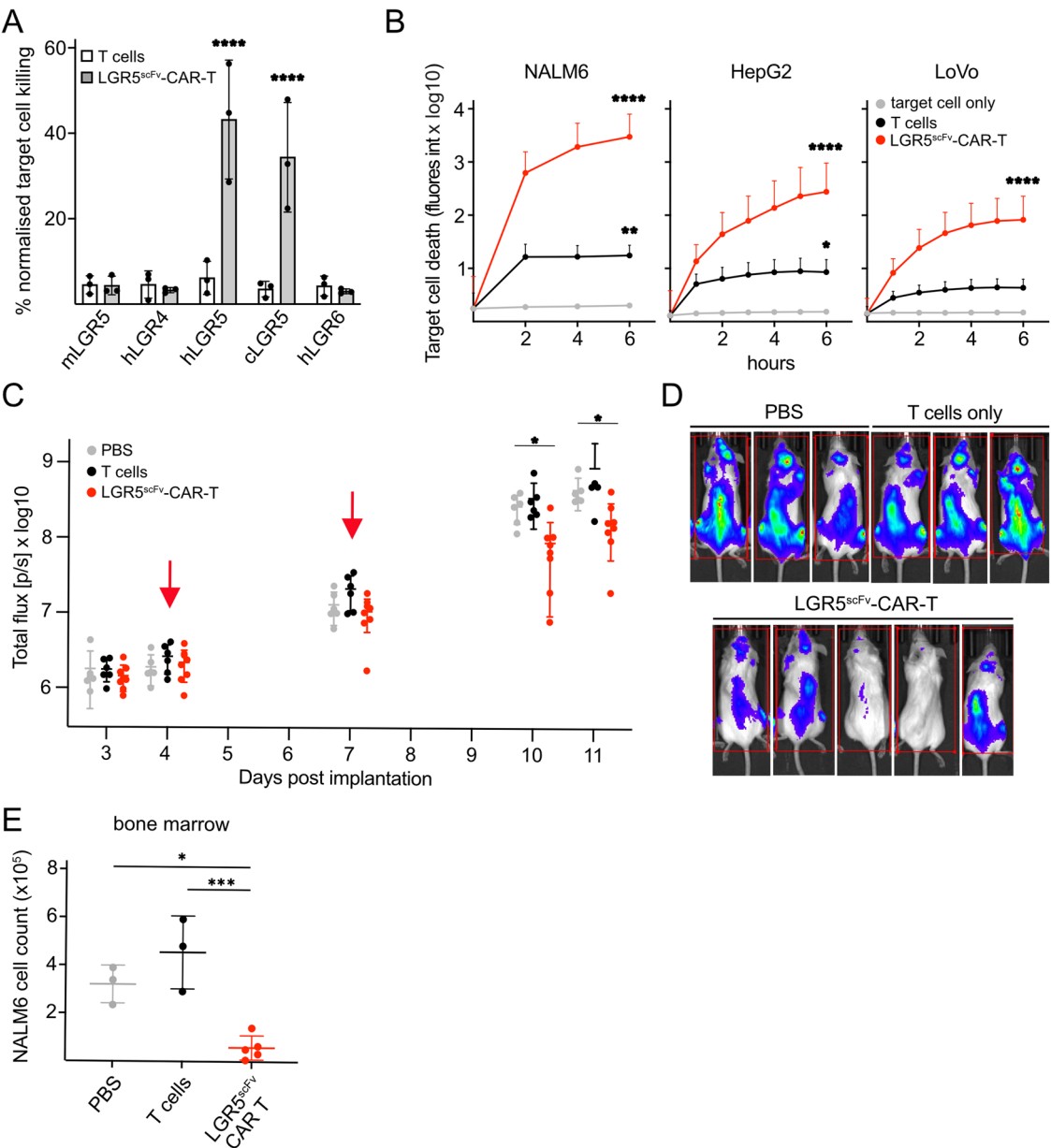

**Figure 7. LGR5^scFv-CAR-T cells specifically kill LGR5+ cancer cells and show pre-clinical efficacy in vivo.**

(A) HEK293T target cells expressing eGFP-fusions of mLgr5, hLGR4, hLGR5, hLGR6, or cLGR5 were incubated with LGR5^scFv-CAR-T cells at an effector-to-target ratio of 10:1 for 9 h. Data are presented as mean expression, +/− SD for treatment with T cells and LGR5^scFv-CAR-T cells generated from three independent healthy donors. (B) Killing kinetics of NALM6, LoVo and HepG2 tumour cells incubated in the presence of mock treatment (no T cells), treatment with non-transduced T cells or LGR5^scFv-CAR-T cells at an effector-to-target ratio of 10:1. Data shown is derived from three independent experiments with 2–6 technical replicates per experiment using T cells from different healthy donors. Data are presented as mean expression, +/− SEM. (C) In vivo efficacy trial evaluating vehicle control treatment (PBS), or treatment with non-transduced T cells or LGR5^scFv-CAR-T cells using the NALM6 tumour model. Mice were injected with 1 × 10^6 NALM6 tumour cells i.v. On days 4 and 7 post implantation (red arrows) tumour-bearing mice were injected i.v. either with PBS or 2.5–5 × 10^6 effector cells (non-transduced T cells or LGR5^scFv-CAR-T cells). Mice were imaged on days 3, 4, 7 and 10 and prior to experimental endpoint on day 11. Data from two independent experiments are presented as mean expression, +/− SD. (D) Representative IVIS images of treated mice on day 10. (E) At the experimental endpoint, NALM6 cells were recovered from one tibia and one femur per mouse and enumerated. Data is representative of three replicate experiments with cells from 3 to 5 mice per treatment group and is presented as mean expression, +/− SD. Source data are available online for this figure.

supports the use of therapeutic molecules such as α-LGR5-ADC, where drug efficacy is independent of immune infiltration.

Apart from its utility as a research tool and biomarker, we show that α-LGR5 is an effective and versatile therapeutic antibody. We confirm the highly dynamic nature of LGR5 protein internalisation and lysosomal trafficking in cancer cells. It is important to note that binding of α-LGR5 to LGR5 does not itself trigger internalisation: the α-LGR5 binding site on LGR5 does not overlap

with the R-spondin binding site, nor does treatment of LGR5-expressing cells with α-LGR5 impact R-spondin potentiation of Wnt pathway activity. Moreover, we observe identical localisation patterns of endogenous LGR5 and fluorescently tagged α-LGR5 in treated cells— puncta distributed throughout the cell body. Our analyses of LGR5 dynamics indicates constitutive internalisation and trafficking to a number of compartments, such as lysosomes.

We compared α-LGR5 internalisation with that of the α-HER2 antibody (Trastuzumab) and show >90% target internalisation of α-LGR5 within 5 min whereas <50% of Trastuzumab was internalised by tumour cells after 3 h. Indeed, α-LGR5-ADCs have been previously road-tested for targeting LoVo CRC cell tumours (Gong et al, 2016; Junttila et al, 2015) as well as an LGR5-expressing ER-PDX tumour model (Hagerling et al, 2020). The present study expands the application of α-LGR5-ADC by targeting NALM6 pre-B-ALL tumours. Our use of two different α-LGR5-ADC treatment regimens, lower than reported by previous studies, has revealed important pharmacodynamic properties of this therapeutic strategy for targeting blood malignancies: the reduction in NALM6 tumour burden manifests up to 4 days post treatment and is only sustained for up to 4 days post treatment at which point we observe tumour regrowth. Our post-tissue census of residual tumour cells at the experimental endpoint indicates that tumour relapse may stem from a reservoir of NALM6 cells within the bone marrow and this tumour compartment can only be effectively targeted by four or more rounds of our α-LGR5-ADC treatment regime. These data are informative for the clinical application of α-LGR5-ADC in targeting blood malignancies and indicate that effective targeting requires increased amounts of α-LGR5-ADC with sustained rounds of treatment. We did not observe off-target toxicities in α-LGR5-ADC-treated mice; however, α-LGR5 does not cross-react with murine Lgr5, and it will be important to evaluate on-target, off-tumour toxicities in an appropriate model. Notably, the previous trials of α-LGR5-ADCs targeting the LoVo tumour model utilising a murine cross-reactive antibody were conducted at greater levels of drug or with prophylactic treatment and no associated treatment toxicities were observed (Gong et al, 2016; Junttila et al, 2015).

Cytotoxic cells of the immune system have formidable characteristics. Targeted secretion of cytotoxic granules at the immune synapse makes the killing very efficient and safe for non-malignant bystanders, and the ability to serially kill many tumour cells dramatically increases efficacy (De La Roche et al, 2016). Here we explore the use of the humanised α-LGR5 scFv fragment in two cell-based modalities. By fusing α-CD3ε$^{scFv}$ to our α-LGR5$^{scFv}$ we generated the CL-BiTE which displayed highly specific and potent activation of human CD4$^+$ and CD8$^+$ T cells and effectively induced cancer cell killing in vitro. In vivo targeting of the pre-B-ALL tumours by treatment with CL-BiTE co-injected with CD8$^+$ T cells led to a significant decrease in tumour burden supporting further development as a novel "off-the-shelf" therapeutic for LGR5+ cancer patients with a functional immune system. We also generated an α-LGR5-CAR-T cellular therapeutic. LGR5$^{scFv}$-CAR-T cells displayed specific and robust tumour cell destruction in vitro and excellent in vivo efficacy. Thus, LGR5$^{scFv}$-CAR-T cells expand the arsenal of potentially effective treatments for LGR5+ liquid and solid cancers.

The primary barrier to he clinical development of α-LGR5 in the three therapeutic modalities are safety studies with a suitable animal model in order to evaluate the potential for on-target/off-tumour effects on stem cell compartments that express LGR5. In this study, we were not able to directly determine the in vivo safety profile of the three α-LGR5-based therapeutic modalities in a mouse model because α-LGR5 recognises human and cynomolgus LGR5 but not the corresponding rodent epitope. Instead, we have carried out extensive in vitro studies in order to establish a therapeutic window for treatment. For instance, we have demonstrated that human tumour cell lines that express higher levels of LGR5 are more vulnerable to all three modalities. Most importantly, we show that the same is true for patient-derived CRC organoids and pre-B-ALL patient cells indicating a favourable safety profile. In the organoid studies, we did not use healthy human colon epithelial organoid controls due to the fact that their culture requires strong Wnt pathway activation via the inclusion of WNT ligands in the media formulation. Wnt pathway activation raises levels of the LGR5 target gene leading to artefactual sensitivity to LGR5-targeting. Future studies will focus on safety and will include cynomolgus studies as we have demonstrated that α-LGR5 shows sufficient cross-reactivity with cLGR5.

Taken together, this study has important implications for cancer research and immune-based therapeutics: (1) our highly specific, versatile α-LGR5 antibody is a particularly useful research tool for determining novel cell biology of human LGR5; (2) LGR5-expressing tumour cells have been validated as bone fide therapeutic targets in CRC, HCC and pre-B-ALL with the possibility of identifying further LGR5-expressing cancers types; and (3) the demonstration that α-LGR5 is an adaptable therapeutic antibody for targeting cancer cells in the ADC, BiTE and CAR-T-cell modalities. While we observed different potencies in vivo for the three modalities in targeting pre-B-ALL tumours, these compatible therapeutic strategies give tremendous scope for accommodating the different pharmacodynamic requirements of various LGR5+ tumour types.

# Methods

**Reagents and tools table**

| Reagent/resource | Reference or source | Identifier or catalogue number |
|---|---|---|
| **Experimental models** | | |
| HEK293T | ECCC | (RRID: CVCL_0063) |
| LoVo | ECCC | (RRID: CVCL_0399), |
| SW480 | ECCC | (RRID: CVCL_0546) |
| HT29 | ECCC | (RRID: CVCL_0320) |
| HCT116 | ECCC | (RRID: CVCL_0291) |
| CaCo-2 | ECCC | (RRID: CVCL_0025) |
| DLD1 | ECCC | (RRID: CVCL_0248) |
| NALM6 | ECCC | (RRID: CVCL_0092) |
| REH | ECCC | (RRID: CVCL_1650) |
| 697 | ECCC | (RRID: CVCL_0079) |
| RS4;11 | ECCC | (RRID: CVCL_0093) |
| HAL-01 | ECCC | (RRID: CVCL_1242) |

| Reagent/resource | Reference or source | Identifier or catalogue number |
|---|---|---|
| NALM16 | ECCC | (RRID: CVCL_1834) |
| SupB15 | ECCC | (RRID: CVCL_0103) |
| KOPN8 | ECCC | (RRID: CVCL_1866) |
| MHH-CALL-2 | ECCC | (RRID: CVCL_1409) |
| NSG, NOD scid gamma strain (*M. musculus*) | Charles River UK Ltd | *NOD.Cg-Prkdc^{scid} Il2rg^{tm1Wjl}/SzJ*, RRID:IMSR_JAX:005557 |
| **Recombinant DNA** | | |
| α-CD19-CAR lentiviral plasmid | Gifted from Dr. John James (University of Warwick) | N/A |
| α-LGR5-CAR lentiviral plasmid | This study | N/A |
| **Antibodies** | | |
| Primary, Secondary Antibodies and Probes | Various manufacturers | Data Appendix Fig. S2 |
| **Oligonucleotides and other sequence-based reagents** | | |
| PCR Primers | This study | Data Appendix Fig. S3 |
| **Chemicals, enzymes and other reagents** | | |
| DMEM ( + D-glucose, L-glutamine, pyruvate) | Gibco | 41966-052 |
| RPMI-1640 ( + L-glutamine) | Gibco | 21875-091 |
| Heat-inactivated FCS | Gibco | 14190-094 |
| Gibson Assembly Cloning Kit | New England Biolabs | E5510S |
| GSH-Sepharose 4B | Cytiva | 17075601 |
| Thrombin protease | Merck | GE27-0846-01 |
| Superdex 75 10/300 gel filtration column | Cytiva | 28-9893-33 |
| DPBS | Gibco | 10010023 |
| GERBU Pä adjuvant | Tebubio | 3111-30 ml |
| Protein G fast flow Sepharose | Cytiva | 17061801 |
| Isopropyl β-D-1-thiogalactopyranoside | Sigma-Aldrich | I5502-5G |
| HiLoad 16/600 Superdex 75 pg column | GE Healthcare | 28989334 |
| Lipofectamine 2000 | Life Technologies | 11668019 |
| Dylight TM650 Antibody labelling kit | Thermo Fisher Scientific | 84535 |
| Dylight TM550 Antibody labelling kit | Thermo Fisher Scientific | 84530 |
| Frag1A peptide | Cambridge Peptide | N/A |
| Phalloidin AF647 | Thermo Fisher Scientific | A30107 |
| Phalloidin AF488 | Thermo Fisher Scientific | A12379 |
| Hoechst 33342 | Life Tech | H3570 |
| Buffy Coats | NHS Blood and Transplant | N/A |

| Reagent/resource | Reference or source | Identifier or catalogue number |
|---|---|---|
| Human leukopaks | Cambridge Bioscience | N/A |
| SepMate PBMC Isolation Tubes | Stemcell Technologies | 85460 |
| Human CD19 Microbeads | Miltenyi Biotec | 130-045-201 |
| Human CD8^+ isolation kit | Miltenyi Biotec | 130-045-201 |
| IgG1 | Sigma-Aldrich | I5451 |
| CellTiter-Glo 2.0 Cell Viability Assay | Promega | G9241 |
| Cultrex extracellular matrix | R&D Systems | 3432-010-01 |
| Qiazol | Qiagen | 79306 |
| Multiscribe Reverse Transcriptase kit | Applied Biosystems | 4311235 |
| SYBR green PCR Master Mix | Applied Biosystems | 4312704 |
| AccuCheck Counting Beads | Thermo Fisher Scientific | PCB100 |
| eflour780 fixable live/dead dye | eBioscience | 65-0865-18 |
| ImmunoCult™ Human CD3/CD28/CD2 T Cell Activator | STEMCELL Technologies | 10970 |
| TexMACS media | Miltenyi Biotec | 130-097-196 |
| Human IL-2 | Miltenyi Biotec | 130-097-746 |
| Penicillin/Streptomycin | Gibco | 15050-063 |
| CellVue membrane dye | Sigma-Aldrich | MINCLARET-1KT |
| TransIT-293 transfection reagent | Geneflow | E7-0026 |
| 0.01% poly-L-ornithine solution | Sigma-Aldrich | A-004-C |
| Apotracker Green | Biolegend | 427403 |
| **Software** | | |
| Imaris software | Bitplane/Oxford Instruments | |
| BD FACSDiva software | BD Bioscience | |
| GraphPad Prism V10.2.3 | https://www.graphpad.com | |
| Arivis Vision 4D software | Arivis | |
| **Other** | | |
| Andor Dragonfly 500 | Oxford Instruments | |
| BD LSR Fortessa | BD Bioscience | |
| BD LSR Symphony | BD Bioscience | |
| PhenoImager HT™ Automated Quantitative Pathology Imaging System | Akoya Biosciences | |
| CLARIOStar | BMG Labtech | |
| IVIS Spectrum in vivo imaging system | Perkin Elmer | |
| Incucyte SX5 | Sartorius | |

## Plasmid constructs

The plasmid for hLGR5-eGFP expression plasmid has been previously described (Snyder et al, 2013). All other LGR transgenes used in the study were constructed by direct replacement of the LGR5 coding sequence with PCR amplicons from the corresponding LGR family coding sequence by either Gibson assembly (New England Biolabs) or restriction enzyme cloning. LGR family coding sequences were sourced as follows: hLGR4 (HG15689), Sino Biological; mLGR4 (MR219497) Origene; mLGR5 (MR219702) Origene; and hLGR6(LGR6_OHu16329D) GenScript.

pGEX-LGR5-NT was created by Gibson assembly of the coding region for amino acids 23–124 of human LGR5 into the pGEX-4T1 bacterial expression vector for purification by GSH-Sepharose 4B (Cytiva) and on-bead cleavage with 2 U of Thrombin protease (Merck) as described by manufacture protocols. The cleaved LGR5 N-terminal fragment was resolved on a Superdex 75 10/300 gel filtration column (Cytiva) equilibrated in PBS using the ÄKTA pure system prior to use as the antigen for α-LGR5 antibody generation. Fragments 1–4 and 1a and 1b were prepared in a similar manner, but without Thrombin cleavage.

Bacterial expression plasmids for RAD display of LGR5 N-terminal fragments (fragments 1–4 and 1a and 1b) were generated by Gibson assembly, for expression in bacteria and purification by heat denaturation and Ni-Sepharose affinity chromatography as previously described (Rossmann et al, 2017).

Sequencing of murine antibody hybridoma clones and generation of transgenic versions of heavy, and light chains of the humanised α-LGR5 antibodies (α-LGR5v4 and α-LGR5v6) and the α-LGR5 scFv fragment were commissioned from Absolute Antibody.

The LC- and CL-BiTEs were generated by Gibson assembly of the coding regions for the ILK-2 signal peptide upstream of the α-CD3ε scFv fragment (clone TR66, amplified from Blinatumomab (Löffler et al, 2000)), and the α-LGR5 scFv fragment into pCDNA3.1. CL-BiTE refers to the fragment order of α-CD3ε scFv N-terminal and the α-LGR5 scFv fragment while LC-BiTE has the scFv fragments in the reverse orientation. For bacterial BiTE expression, the CL-BiTE, without the ILK-2 signal peptide, was transferred by Gibson assembly into the pET-Duet vector, containing the E. coli disulphide bond isomerase, DsbC (at the secondary insertion site).

The LGR5scFv-CAR was generated from a α-CD19-CAR lentiviral plasmid (a kind gift from Dr John James, University of Warwick (James, 2018)) that contained the coding region for the α-CD19 scFv domain (clone FMC63) fused in turn to the CD8 stalk and transmembrane regions, the CD28 secondary co-stimulatory domain, and mScarlet. The α-LGR5 scFv fragment was used to replace the α-CD19 scFv domain in the α-CD19-CAR lentiviral plasmid using Gibson Assembly.

All constructs generated were verified by Sanger or Nanopore sequencing and primer sequences are provided in Appendix Table S3.

## Antibodies and probes

Commercially sourced antibodies and probes are listed in Appendix Table S2.

## Mammalian cell lines

Cell lines were purchased from the European Collection of Cell Cultures (ECCC) and have been authenticated by short tandem repeat (STR) DNA profiling. Upon receipt, cell lines were frozen, and individual aliquots were taken into culture, typically for analysis within <10 passages. Cells were grown in a humidified incubator at 37 °C and 5% $CO_2$ and tested mycoplasma negative (MycoProbe® Mycoplasma Detection Kit, R&D systems). HEK293T cells (RRID: CVCL_0063), and the colorectal cancer cell lines LoVo (RRID: CVCL_0399), SW480 (RRID: CVCL_0546), HT29 (RRID: CVCL_0320), HCT116 (RRID: CVCL_0291), CaCo-2 (RRID: CVCL_0025) and DLD1 (RRID: CVCL_0248), were maintained in DMEM medium supplemented with 10% heat-inactivated FCS (Gibco). pre-B-ALL cell lines NALM6 (RRID: CVCL_0092), REH (RRID: CVCL_1650), 697 (RRID: CVCL_0079), RS4;11 (RRID: CVCL_0093), HAL-01 (RRID: CVCL_1242), NALM16 (RRID: CVCL_1834), SupB15 (RRID: CVCL_0103), KOPN8 (RRID: CVCL_1866), and MHH-CALL-2 (RRID: CVCL_1409) were maintained in RPMI-1640 (Gibco) medium supplemented with 10% heat-inactivated FCS (Gibco).

## Generation of α-LGR5 antibodies, α-LGR5 scFv fragment and BiTE molecules

The somatic fusion was made between the SP2 myeloma cell line and splenocytes from NMRI mice or SPRD rats (Taconic) that were SC immunised twice, at a 14-day interval, with 30 mg of the LGR5 N-terminal fragment glutaraldehyde-coupled to diphtheria toxoid. The antigen was administered with the GERBU Pä adjuvant according to the manufacturer's recommendations. Four days prior to the fusion, the animals received an I.V. injection boost of 15 mg antigen administered with adrenaline.

Purification of immunoglobin fractions was carried out by absorption of 1 L of hybridoma supernatant, to a 3 ml packed volume of Protein G fast flow Sepharose equilibrated in PBS. After extensive washing of the columns with PBS, the immunoglobin fraction was eluted with 100 mM glycine, pH 2.7 and immediately neutralised with 200 mM Tris pH 8.0.

Production of α-LGR5v4 and α-LGR5v6 was carried out by transfection of HEK293T ($4 \times$ T175 flasks) with 15 µg of each encoding plasmid per flask. Approximately 200 ml of conditioned was collected at 2- and 4 days post-transfection and antibodies were purified by Protein G chromatography, as above.

The α-LGR5 scFv fragment and the CL- and LC BiTEs were produced from expression in HEK293T cells ($2 \times$ T175 flasks) transfected with 20 µg of the encoding plasmids. Conditioned media containing antibodies was collected at 2- and 4 days post-transfection. The α-LGR5 scFv fragment was purified by Ni/NTA chromatography and the CL- and LC BiTEs were purified by FLAG-Agarose chromatography (Sigma-Aldrich) according to the manufacturer's protocols.

For large-scale production of CL-BiTE for in vivo studies, the CL-BiTE-coding regions were subcloned into the pET-Duet vector containing the E. coli disulphide bond isomerase, DsbC (at the secondary insertion site) using Gibson assembly. pET-CL-BiTE was transferred to the host E. coli strain SHuffle T7 and after 16 h expression at 16 °C with 1 mM isopropyl β-D-1-thiogalactopyranoside cells were pelleted and the CL-BiTE protein was purified by Ni/NTA chromatography followed by size exclusion chromatography on a HiLoad 16/600 Superdex 75 pg column (GE Healthcare). The resolved protein was concentrated and transferred to PBS by buffer exchange using a PD-10 column.

## Cellular assays and manipulation

Indirect immunofluorescence, western blot analysis, flow cytometry, TopFlash assays and quantitative real-time PCR have been previously described (Chen et al, 2019; De La Roche et al, 2014). For immunofluorescence, detection was carried out by confocal spinning disc microscopy using an Andor Dragonfly 500 (Oxford Instruments). Images were processed using Imaris software (Bitplane/Oxford Instruments). Flow cytometry was carried out with a BD LSR Fortessa or BD LSR Symphony cell analyzer using the BD FACSDiva software (BD Biosciences Inc.). Quantitative real-time PCR (qRT-PCR) has been previously described (6) and used Taqman probes specific for human LGR5 (Life Technology, Hs00969422_m1) and for TBP (Life Technology, Hs00427620_m1) as control housekeeping gene. The Prism software package was used to graph data sets from TopFlash assays and qRT-PCR experiments and for statistical analysis using two-tailed Student's $t$ test.

For overexpression of LGR family proteins, HEK293T cells were transfected with the corresponding plasmids using lipofectamine 2000 (Life Technologies) according to the manufacturer's recommendations. Cells were transfected overnight and recovered in culture media for an additional 16 h prior to immunofluorescence, western blot or flow cytometry.

The fluorescent versions of α-LGR5 (Fl-α-LGR5), α-LGR5v4 (Fl-α-LGR5v4) and α-LGR5v6 (Fl-α-LGR5v6) were generated using the Dylight TM650 Antibody labelling kit (Thermo Fisher Scientific). Fluorescent Trastuzumab (Fl-α-HER2) was generated using the Dylight TM550 Antibody labelling kit, respectively. In some controls for flow cytometry and immunofluorescence, Fl-α-LGR5 was pre-incubated with supra-stoichiometric amounts of RAD-displayed Frag1A or Frag1B, or the Frag1A peptide (Cambridge Peptides) at a molar ratio of 10:1.

Antibody internalisation assays were carried out by incubation of cells with 20 µg/ml of Fl-α-LGR5 or Fl-α-HER2 for various timepoints, followed by fixation in 4% paraformaldehyde and immunofluorescence. Images derived from z-stacks of cells were analysed for Fl-α-LGR5v4 and fluorescent signals from antibodies against various cellular markers, fluorescent phalloidin to visualise cortical actin, and Hoechst 33342 to visualise nuclei. Images were processed and analysed used Arivis Vision 4D software. Image segmentation to delineate fluorescent features: whole cells, delineated by cortical F-actin, nuclei and puncta) in 3D was carried out using Blob Finder. The Arivis Vision 4D software was then used to classify puncta associated with the cell membrane or within the cells and to determine the degree of co-localisation between LGR5 and compartment-specific markers—co-localisation is defined as >50% overlap with LGR5 puncta.

For the internalisation kinetics experiments, images were analysed using Arivis Vision 4D software, combined with a deep-learning segmentation using Cellpose. Two regions were defined: cell outer membrane, and cell cytoplasm. Segmented dotted signal corresponding to Fl-α-LGR5 or Fl-α-LGR5v4, as well as segments-like signal corresponding to Fl-α-HER2, were classified according to their location within these two regions denoted as 'associated' or 'internalised'. Ratiometric analysis was performed to quantify the internalisation of both markers at various timepoints.

## TCGA data mining for LGR5 expression in cancers

Publicly available gene expression data (RNAseq V2) from The Cancer Genome Atlas (TCGA; https://www.cancer.gov/about-nci/organization/ccg/research/structural-genomics/tcga) were downloaded using Firebrowse (http://firebrowse.org/). Gene-level read counts were quantile normalised using Voom (Law et al, 2014) and (log2 median-centred) LGR5 gene expression was determined for each sample. Tumour subtypes for which more than 70% of samples had higher than pan-cancer median LGR5 expression were triaged as "high LGR5 tumours".

## Patient samples and immunofluorescence detection of LGR5 and β-catenin

All human tissue biopsies used in the study were paraformaldehyde-fixed paraffin-embedded and probed with α-LGR5 and an antibody to β-catenin and visualised by fluorescent secondary antibodies to mouse labelled with Alexa 488 and rabbit labelled with Alexa 555. All immune-stained samples were imaged using the PhenoImager HT™ Automated Quantitative Pathology Imaging System (Akoya Biosciences). Scoring of all human biopsies for simultaneous LGR5 and β-catenin expression was performed by an individual blind to the provenance of the samples and graded from no expression—0 to high expression—3 for all sample sets. The Prism software package was used for plotting of LGR5 or β-catenin expression levels for all biopsy sample sets and for determining statistical differences from healthy tissue using two-tailed Students $t$ test. Unless otherwise notes, all relevant legal and ethical guidelines of the Addenbrooke's Hospital (Cambridge, UK) were followed for collection of samples and provision for the present study. Informed consent for research application was obtained from all subjects.

LGR5 expression in individual colorectal cancer cases and adjacent healthy tissue were determined for biopsies provided by Dr Olivier Giger (OG; (IRAS: 162057). Within these tissue samples, regions were annotated as normal, dysplastic or invasive tissue from consecutive H&E sections. LGR5 and β-catenin protein levels were additionally determined in CRC by immunofluorescence using the Bern CRC sample set, provided by Dr Inti Zlobec. The Bern CRC sample set is a highly annotated tumour microarray (TMA) consisting of 160 individual cases in duplicate with determined phenotypic feature—gender, age, tumour stage, therapeutic intervention and MSI status. Biopsies used in the construction of the TMA were collected under ethics 2020-00498 granted by the Ethical Committee of the Canton of Bern, Switzerland. All relevant guidelines of the Institute of Pathology, University of Bern, Canton of Bern, Switzerland were followed for construction of the TMA.

The Cambridge HCC TMA consists of 104 human liver samples and was collected by Drs Sarah Aitken and Matthew Hoare with informed consent from Addenbrooke's Hospital, Cambridge, UK, according to procedures approved by the East of England Local Research Ethics Committee (16/NI/0196 and 20/EE/0109). Liver samples classified as healthy were obtained from resections from females with inflammatory adenoma of the liver (2 individuals) or focal nodular hyperplasia (2 individuals) or from a male with an HNF1α-inactivated adenoma. All biopsies of healthy liver tissue were taken from patients between the ages of 25 and 36.

High-grade serous ovarian carcinoma (HGSOC) samples comprising the Cambridge ovarian cancer TMA were provided by Prof James Brenton. Tumour samples were obtained from patients enrolled in the Cambridge Translational Cancer Research Ovarian Study 04 (CTCROV04, short OV04) study approved by the Institutional Ethics Committee (08/H0306/61). Samples were processed following standardised operating protocols as outlined in the OV04 study design. Tissue quality was assessed using haematoxylin and eosin (H&E) sections, and high-purity regions were selected for tissue microarray (TMA) generation (using 0.1-cm cores). The TMA consisted of a healthy fallopian tube (FT; 27 samples), 28 ovarian cancer cases (OvC) and 14 omentum cancer cases (OmC).

The Cambridge brain cancer TMA consists of five samples of healthy brain tissue, five from low-grade glioma and five from glioblastoma that were collected via the ICARUS biorepository, Addenbrooke's Hospital, according to approved local research ethics (18/EE/0172).

Sections of PDAC and healthy pancreas were provided by Dr. Eva Serrao and were obtained from the Cambridge University Hospital Human Tissue bank, Cambridge, UK, according to procedures approved by the Cambridge South Research Ethics Committee (18/EE/0227).

Primary haematological malignancy samples used in this study were provided by Blood Cancer UK Childhood Leukaemia Cell Bank (now VIVO Biobank) under ethics—16/SW/0219 and Cambridge Blood and Stem Cell Biobank under ethics—18/EE/0199.

All experiments conform to the principles set out in the WMA Declaration of Helsinki and the Department of Health and Human Services Belmont Report.

Buffy Coats from healthy donors were acquired from NHS Blood and Transplant (Cambridge, ethics 17/YH/0304) or as fresh human leukopaks (Cambridge Bioscience). PBMCs were isolated using SepMate PBMC Isolation Tubes (Stemcell Technologies) and B cells and CD8+ T cells were isolated from these using the human CD19 Microbeads (Miltenyi Biotec) or the human CD8+ isolation kit (Miltenyi Biotec).

## Immunohistochemistry

Tissues were fixed in neutral buffered formalin (Sigma #HT5014) for 24 h and transferred to 70% EtOH for 24 h. Following fixation, the tissues were processed overnight, embedded in paraffin and sectioned at 3 μm using a Leica microtome. Following baking, sections were dewaxed and rehydrated on Leica's automated ST5020. Antibody staining was run on Leica's automated Bond-III platform, in conjunction with their Polymer Refine Detection System (DS9800). Antigen retrieval was achieved with Tris EDTA (Leica's Epitope Retrieval Solution 2 #AR9640), incubated for 20 min at 100 °C. Protein block (Dako #X090930-2) and DAB Enhancer (Leica #AR9432) were applied to all sections. Antibodies and conditions are listed below:

| Target | Catalogue No. | Dilution/conc. | Retrieval | Modifications |
| --- | --- | --- | --- | --- |
| CD20 | Novocastra, NCL-L-CD20-L26 (CD20-L26-L-CE) | 0.95 μg/ml | Tris EDTA, 20' | Protein Block, DAB Enhancer |
| LGR5 | Murine clone 2 | 1:1500 | Tris EDTA, 20' | Protein Block, MOM block, DAB Enhancer |

## ADC generation and in vitro killing assays

All antibodies and IgG1 (Sigma-Aldrich) were coupled to MMAE through a divinyl pyrimidine bridging linker inserted within the interchain disulphide linkages for the precise drug-to-antibody ratio of 4 (Walsh et al, 2019). For in vitro killing assays, LoVo target cells were seeded into opaque 96-well plate for overnight, allowing settlement before treatment using ADCs with cleavable linker or non-cleavable control linker at doses of 30, 10, 3, 1, 0.3 and 0.1 nM for 3 days. NALM6 and REH target cells were seeded into opaque 96-well plate and treated directly by ADCs with cleavable linker or non-cleavable control linker at doses of 30, 10, 3, 1, 0.3 and 0.1 nM for 3 days. On day 3, cell viability was evaluated by CellTiter-Glo 2.0 Cell Viability Assay (Promega) according to the manufacturer's instructions. Bioluminescence was measured using a CLARIOStar (BMG Labtech).

## Organoid cultures

Organoid models were derived from biopsies of surgical resections obtained from Cambridge University Hospital Human Tissue bank, Cambridge, UK under ethics 15/WA/0131 according to Sato et al (Sato et al, 2011). All organoid models were cultured in 20 μl domes of Cultrex extracellular matrix (R&D Systems) in wells of a 48-well plate using a defined media formulation described by Urbischek et al, (Urbischek et al, 2019).

For determining LGR5 gene expression in the organoid models, RNA was produced from 2 wells (approximately 400 individual organoids) using Qiazol (Qiagen). cDNA generation was performed using the Multiscribe Reverse Transcriptase kit (Applied Biosystems). Primer pairs to quantify LGR5 and the housekeeping gene TBP were: 5'-GAGTTACGTCTTGCGGGAAAC (forward) and 5'-TGGGTACGTGTCTTAGCTGATTA (reverse), 5'-CCCGAAACGCCGAATATAATCC (forward) and 5'-AATCAGTGCCGTGGTTCGTG (reverse). Quantitative PCR reactions were carried out using SYBR green PCR Master Mix (Applied Biosystems).

## Mouse strain for experimentation

Immunodeficient NSG mice (NOD scid gamma; strain *NOD.Cg-Prkdc^{scid} Il2rg^{tm1Wjl}/SzJ*, RRID:IMSR_JAX:005557) were purchased from Charles River UK Ltd (Margate, UK) and housed under specific pathogen-free conditions at the University of Cambridge, CRUK Cambridge Institute in accordance with UK Home Office regulations under PPL: PP0228268. Where applicable, experimentation with mice adhered to Arrive guidelines. Experimentation was carried out using female and male mice between 6 and 20 weeks of age. Where appropriate individuals processing samples were blind to the provenance of the experimental group identifiers.

## In vivo performance of the murine and humanised α-LGR5-ADCs

NALM6 cells were transduced with lentivirus encoding a LucEYFP reporter (a kind gift from Prof. Kevin Brindle; NALM6-LucYFP cells).

NSG mice were injected intravenously (iv) with NALM6-LucYFP cells by tail vein injection and monitored for weight loss and imaging using the IVIS Spectrum in vivo imaging system

(Perkin Elmer) in a 2–3-day interval. For IVIS imaging, mice were given D-luciferin at a dose of 150 mg/kg by intraperitoneal injection 10 min prior to imaging under general anaesthesia with isoflurane. Mice were randomised into control and treatment groups according to the tumour load determined by the bioluminescence signal detected on day 5 post-initiation (PI). Treatment with mIgG-ADC or α-LGR5-ADC at a dose of 5 mg/kg was carried out by iv injection starting on day 6 for 4 times every other day. Treatment with α-LGR5v6-ADC (control) or α-LGR5v4-ADC at dose of 5 mg/kg was carried out by iv injection on days 6 and 8. At the experimental endpoint on day 20, all animals were euthanised in accordance with Schedule 1 of the Animals (Scientific Procedures) Act 1986. Spleen, blood, heart, kidney, lung, liver, small intestine, femur and tibia were collected at the experimental endpoint for histology and/or flow cytometry analysis.

Single-cell suspensions prepared from spleen, blood, and bone marrow of femur and tibia were stained with eflour780 fixable live/dead, followed by fluorescence-conjugated antibody against human CD19 for identification of NALM6-LucYFP cells (CD19$^+$ EYFP$^+$) by flow cytometry analysis. Absolute number of NALM6-LucYFP cells was determined using AccuCheck Counting Beads (Cat# PCB100, Thermo Fisher Scientific).

## In vitro T-cell activation and target cell-killing assays

PBMCs used for BiTE assays were obtained as fresh human leukopaks (Cambridge Bioscience) or as buffy coats obtained from NHS Blood and Transplant under ethics 17/YH/0304.

In vitro T-cell activation assays were initiated with the addition of 4 nM CL or LC BiTEs or α-LGR5$^{scFv}$ as a negative control to a mixture of $1 \times 10^6$ PBMCs and $1 \times 10^6$ NALM6-LucYFP cells. After 24-h incubation, cells were stained with eflour780 fixable live/dead dye (eBioscience) and fluorescence-conjugated antibodies against CD4, CD8, CD25 and CD69 for determination of the T-cell differentiation and activation markers by flow cytometry.

BiTE-mediated killing assays were carried out using cytotoxic CD8$^+$ T cells isolated from PBMCs, as described, and stimulated for 72 h with 25 μL/mL ImmunoCult™ Human CD3/CD28/CD2 T Cell Activator (STEMCELL Technologies). CD8$^+$ T cells were cultured in TexMACS media (Miltenyi Biotec) supplemented with 100 U/ml human IL-2 (Miltenyi Biotec) and 100 U/ml Penicillin/Streptomycin (Gibco). Cells were restimulated at $1 \times 10^6$ cells/mL concentration for 48 h on day 8–10 with 1 μg/mL plate bound anti-CD3ε antibody (300438 clone UCHT1, Biolegend). BiTE-mediated killing assays were carried out on day 14–16 of culture using the CL-BiTE at a concentration of 4 nM. Target cells were labelled with CellVue membrane dye (Sigma-Aldrich) according to the manufacturer's instructions, and killing was assessed after 6–9 h of co-culture at 37 °C. At the endpoint, cells were labelled with eFluor780 fixable live/dead dye (Thermo Fisher) and analysed for viability by flow cytometry.

## Generation of LGR5$^{scFv}$-CAR-T and LGR5$^{scFv}$-CAR-NK cells

The lentivirus for LGR5$^{scFv}$-CAR expression was produced in HEK293T cells by transfection of the lentiviral plasmid, lentiviral packaging plasmid pCMV-dR8.91 and lentiviral envelope plasmid pMD.G using TransIT-293 transfection reagent (Geneflow). The viral supernatant was harvested at 48 h and 72 h and concentrated by ultracentrifugation.

**The paper explained**

**Problem**
There is an urgent clinical need for effective therapies targeting CRC, HCC and pre-B-ALL cancer cells. LGR5 is highly expressed by these malignancies and—as a cell surface protein— serves as an excellent candidate for novel immunotherapeutics. However, there are currently no LGR5-directed immunotherapeutics approved owing to the paucity of validated antibodies.

**Results**
We have developed a novel, highly specific antibody to human and primate LGR5 that we have validated as a research tool and for diagnostic detection of LGR5 in CRC, HCC and pre-B-ALL. We have further developed the LGR5 antibody into ADC-, BiTE- and CAR-based therapeutic modalities. In vivo studies indicate effective targeting of pre-B-ALL tumours.

**Impact**
Our study establishes a versatile α-LGR5 antibody and a portfolio of effective immunotherapeutics targeting LGR5-expressing malignancies.

NK92 cells were transduced with an MOI of 10 and after 48–72 h mScarlet-expressing cells were sorted by fluorescence-activated cell sorting (FACS). CD8$^+$ T cells isolated from PBMCs were transduced with an MOI of 5 and LGR5$^{scFv}$-CAR expressing cells isolated by FACS based on mScarlet fluorescence.

## In vitro LGR5$^{scFv}$-CAR-NK and LGR5$^{scFv}$-CAR-T-cell-killing assays

HEK293T cells overexpressing hLGR5-eGFP, hLGR5-eGFP, hLGR6-eGFP, mLGR5-eGFP or cLGR5-eGFP used as target cells in VITAL killing assay and were pre-loaded with CellVue membrane dye (Sigma-Aldrich) and LGR5$^{scFv}$-CAR-NK cells or LGR5$^{scFv}$-CAR-T cells added at variable effector to target cell ratios. Target cell killing was determined by flow cytometry.

Killing assays using the adherent tumour cell lines LoVo and HepG2 cells were carried out by seeding on flat-bottom 96-well plates. NALM6-LucYFP cells were seeded onto 0.01% poly-L-ornithine solution (Sigma-Aldrich) coated plates. LGR5$^{scFv}$-CAR-NK cells or LGR5$^{scFv}$-CAR-T cells were added at variable effector to target ratios and cell death was assessed over 6 h using Apotracker Green (Biolegend; 1:200 dilution) in an Incucyte SX5 (Sartorius).

## In vivo targeting of NALM6 tumours with CL-BiTE or α-LGR5-CAR-T treatment cells

In vivo tumour targeting experiments were initiated in NSG mice by injection of $1 \times 10^6$ NALM6-LucYFP cells. At 3 days PI, mice were allocated into treatment groups based on tumour load determined via IVIS imaging. On day 4 and day 7 PI treatments were carried out with either 100 μg CL-BiTE and $7$–$10 \times 10^6$ CD8$^+$ T cells or LGR5$^{scFv}$-CAR-T cells. At the experimental endpoint on day 11, all animals were euthanised and spleen, blood, femur and tibia were collected for analysis.

Single-cell suspensions prepared from spleen, blood and bone marrow were stained with eFlour780 fixable live/dead, followed by

fluorescence-conjugated antibody against human CD19 for identification of NALM6-LucYFP cells (CD19 + YFP +), and against human CD8 for identification of injected T cells by flow cytometry. The absolute number of NALM6-LucYFP cells was determined using AccuCheck Counting Beads (Thermofisher).

## Statistical tests on data

Tests on data sets for statistical significance are indicated at the *$P < 0.05$, **$P < 0.01$, ***$P < 0.001$, ****$P < 0.0001$ levels. Detailed information about the statistical tests conducted and exact $P$ values obtained can be found in Appendix Table S4.

# Data availability

This study includes no data deposited in external repositories.

The source data of this paper are collected in the following database record: biostudies:S-SCDT-10_1038-S44321-024-00121-2.

# Peer review information

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

## Acknowledgements

Special thanks and gratitude go to the BRU core facility (CRUK Cambridge Institute) for excellent animal husbandry, the Preclinical Imaging Core (CRUK Cambridge Institute) and especially Aude Vernet for help with in vivo imaging, the Histopathology core (CRUK Cambridge Institute) and especially Jodi Miller for help with establishing the IHC protocol, the Light Microscopy core (CRUK Cambridge Institute), the Compliance and Biobanking core (CRUK Cambridge Institute) and Flow Cytometry core (CRUK Cambridge Institute). We would like to acknowledge Prof. Martin D. Berger from the Department of Medical Oncology at the University Hospital Bern for support with clinical data. This work was supported by Cancer Research UK (MdlR (A22257) that includes PhD studentships to NM and FB, a Sir Henry Dale Fellowship jointly funded by the Wellcome Trust and the Royal Society (MdlR [107609/Z/15/Z]), a Wellcome Trust Discovery Award (MdlR [227432/Z/23/Z]), Wellcome Institutional Translational Partnership Award (MAD, 222062/Z/20/Z). SER is supported by a Clinician Scientist Fellowship from Cancer Research UK (C67279/A27957). Research in the Wellcome—MRC Cambridge Stem Cell Institute is funded by a grant from the Wellcome Trust (203151/Z/16/Z). SJA is supported by a Clinician Scientist Fellowship from Cancer Research UK (RCCCSF-May23/100001), Core funding from the Medical Research Council (RG94521), and the Pathological Society (3903171). IO is supported by a Cancer Research UK Pioneer Award (C63389/A30462). MH is supported by a CRUK Clinician Scientist Fellowship (C52489/A19924) and a CRUK Accelerator Award (C18873/A26813).

## Author contributions

**Hung-Chang Chen**: Investigation; Characterised the α-LGR5 antibody through in vitro and in vivo validation of the α-LGR5-ADC and contributed to manuscript writing. **Nico Mueller**: Formal analysis; Investigation; Writing—review and

editing; α-LGR5 antibody characterisation in vitro, created the CAR constructs and performed all experiments to validate the α-LGR5-CAR-T and α-LGR5-NK cells in vitro and determined pre-clinical efficacy in vivo. Contribution to manuscript writing. Prominent contribution to experiments for manuscript revision. **Katherine Stott**: Investigation; In vitro binding measurements to determine affinities of α-LGR5 antibodies, ADCs and the scFV fragment to the LGR5 epitope. **Chrysa Kapeni**: Investigation; Contribution to in vivo studies for α-LGR5-ADC and carried out all in vivo studies with α-LGR5-BiTE. **Eilidh Rivers**: Investigation; Created the RAD display constructs, generated protein and carried out the α-LGR5 epitope mapping experiments. **Carolin M Sauer**: Investigation; Analysed LGR5 transcript expression using TCGA data sets. **Flavio Beke**: Investigation; Performed in vitro activation and killing assays for the α-LGR5-BiTE molecule. **Stephen J Walsh**: Investigation; Design and production of Antibody-Drug conjugates. **Nicola Ashman**: Resources; Methodology; Contribution to the production of α-LGR5-ADCs. **Louise O'Brien**: Investigation; qRT-PCR experiments and general support for experimentation in the MdlR laboratory. **Amir Rafati Fard**: Investigation; Carried out analyses of LRG5 protein expression in CRC biopsies. **Arman Ghodsinia**: Investigation; Created the LC- and CL-BiTE expression vectors and established protein production pipeline. **Changtai Li**: Resources; Investigation; Cultured the CRC organoid models, carried out transcript and protein expression analysis and performed killing assays. **Fadwa Joud**: Formal analysis. **Olivier Giger**: Formal analysis; Histopathology and anaysis of LRG5 protein expression in CRC biopsies. **Inti Zlobec**: Resources; Formal analysis; Assembly, annotation and provision of the Bern CRC TMA. **Ioana Olan**: Resources; Formal analysis; Performed correlation analysis of LGR5 expression in HCC and established phenotypic metrics for disease subtyping. **Sarah J Aitken**: Resources; Formal analysis; Assembly, annotation and provision of HCC and healthy human liver microarrays. **Matthew Hoare**: Resources; Assembly, annotation and provision of HCC and healthy human liver microarrays.

**Richard Mair**: Resources; Assembly, annotation and provision of TMA samples; healthy human brain, LGG and GBM. **Eva Serrao**: Resources; Provided tissue slides of healthy human pancreas and PDAC. **James D Brenton**: Resources; Assembly, annotation and provision of tissue/tumour microarray containing healthy human fallopian tube and ovarian/omentum malignant tissues. **Alicia Garcia-Gimenez**: Resources; Methodology; In vivo expanded and provided human pre-B-ALL samples. **Simon E Richardson**: Resources; Methodology; Characterisation, annotation and provision of human pre-B-ALL samples. **Brian Huntly**: Resources; Methodology; Provision of human pre-B-ALL samples. **David R Spring**: Resources; Methodology; Supervision of α-LGR5-ADC production and provision of ADC linker technology. **Mikkel-Ole Skjoedt**: Resources; Methodology; Generated monoclonal antibodies against human LGR5. **Karsten Skjødt**: Methodology; Generated monoclonal antibodies against human LGR5. **Marc de la Roche**: Conceptualisation; Data curation; Funding acquisition; Investigation; Writing—original draft; Writing—review and editing; Conceived the project, directed research, generated monoclonal antibodies against human LGR5, designed the experiments, performed experiments, analysed results and wrote the manuscript. **Maike de la Roche**: Conceptualisation; Supervision; Funding acquisition; Investigation; Writing—original draft; Project administration; Writing—review and editing; Conceived the project, directed research, generated monoclonal antibodies against human LGR5, designed the experiments, performed experiments, analysed results and wrote the manuscript.

Source data underlying figure panels in this paper may have individual authorship assigned. Where available, figure panel/source data authorship is listed in the following database record: biostudies:S-SCDT-10_1038-S44321-024-00121-2.

## Disclosure and competing interests statement

The authors declare no competing interests. The LGR5 antibodies in the manuscript and all commercial use fall under patent filing PCT/GB2023/050512, inventors—Maike de la Roche and Marc de la Roche.

# Expanded View Figures

**Figure EV1.  Specificity of LGR5 antibodies generated in the study.**  ▶

(**A**) Amino acid sequence of the human LGR5 antigen used for mouse immunisation and generation of α-LGR5; numbering starts at Gly1 in the processed human LGR5 (hLGR5), lacking the signal sequence. The sequence is annotated with the Fragments used in the RAD display experiments that map the α-LGR5 epitope to Frag1A. Fragments do not cover sequences that match the murine Lgr5. *Below* – location of the antigenic region (in red) within the structure of the extracellular domain of LGR5 (atomic coordinates for the model taken from (Peng et al, 2013). (**B**) Configuration of the LGR family transgenic constructs used in the study. All expressed LGR proteins contain a common N-terminal hemagglutinin (HA) tag and fusion at the C-terminus to the vasopressin V2 receptor C-terminal tail (V2R) followed by eGFP. (**C**) Western blot analysis of HEK293T lysates expressing the murine LGR5 (mLGR5), the human LGR5 (hLGR5) and the *cynomolgus* LGR5 (cLGR5) probed with α-LGR5 hybridoma clones 1, 3 and 4 and antibodies to HA and vinculin, as noted. No specific immune reactivity was observed when probing the western blots with the other 14 hybridoma clones. (**D**) Sequence conservation amongst the α-LGR5 hybridoma clones within the complementary determining regions (CDRs). Conserved amino acids relative to α-LGR5 clone 1 for clones 2–4 are represented by a dash. Amino acid differences are indicated "X". (**E**) Western blot analysis of the Fragments delineated above (Fig. EV1A) as RAD-displayed fusion peptides using α-LGR5 hybridoma clones 1 (top panel), 3 (middle panel) and 4 (bottom panel). (**F**) Sequence alignment of the N-terminal 15 amino acids of human LGR5, corresponding to Frag1A, with the corresponding region in the other LGR family members. Sequences were aligned based on three invariant cysteine residues denoted by asterisks. The amino acid difference in the *cynomolgus* sequence is underlined. (**G**) Wnt pathway reporter assays (TopFlash assays) for HEK293T cells transfected with either eGFP or human LGR5-eGFP (hLGR5-eGFP), treated with Wnt3A ligand, R-spondin and either IgG1 or α-LGR5 at levels of approximately 10-fold molar excess over Wnt3A ligand. ns, no significant difference. Data is presented as mean expression, $+/-$ SD for 3 biological replicates. *ns*, no significant difference in Wnt pathway reporter activity. (**H**) Immunofluorescent detection of HEK293T cells expressing transgenic LGR4-eGFP or LGR5-eGFP (*left panels*, green) using Fl-α-LGR5 (*middle panels*, red). *Right panels* merged fluorescent signals. Scale bars, 10 μm. (**I**) Flow cytometric analysis of HEK293T cells expressing mLGR4-eGFP (*top left*), mLGR5-eGFP (*top right*) and hLGR5-eGFP (*bottom panels*) using Fl-α-LGR5. For analysis of the hLGR5-eGFP expressing HEK293T cells, Fl-α-LGR5 was pre-incubated with either RAD-Frag1A or RAD-Frag1B (*bottom left and right*). Source data are available online for this figure.

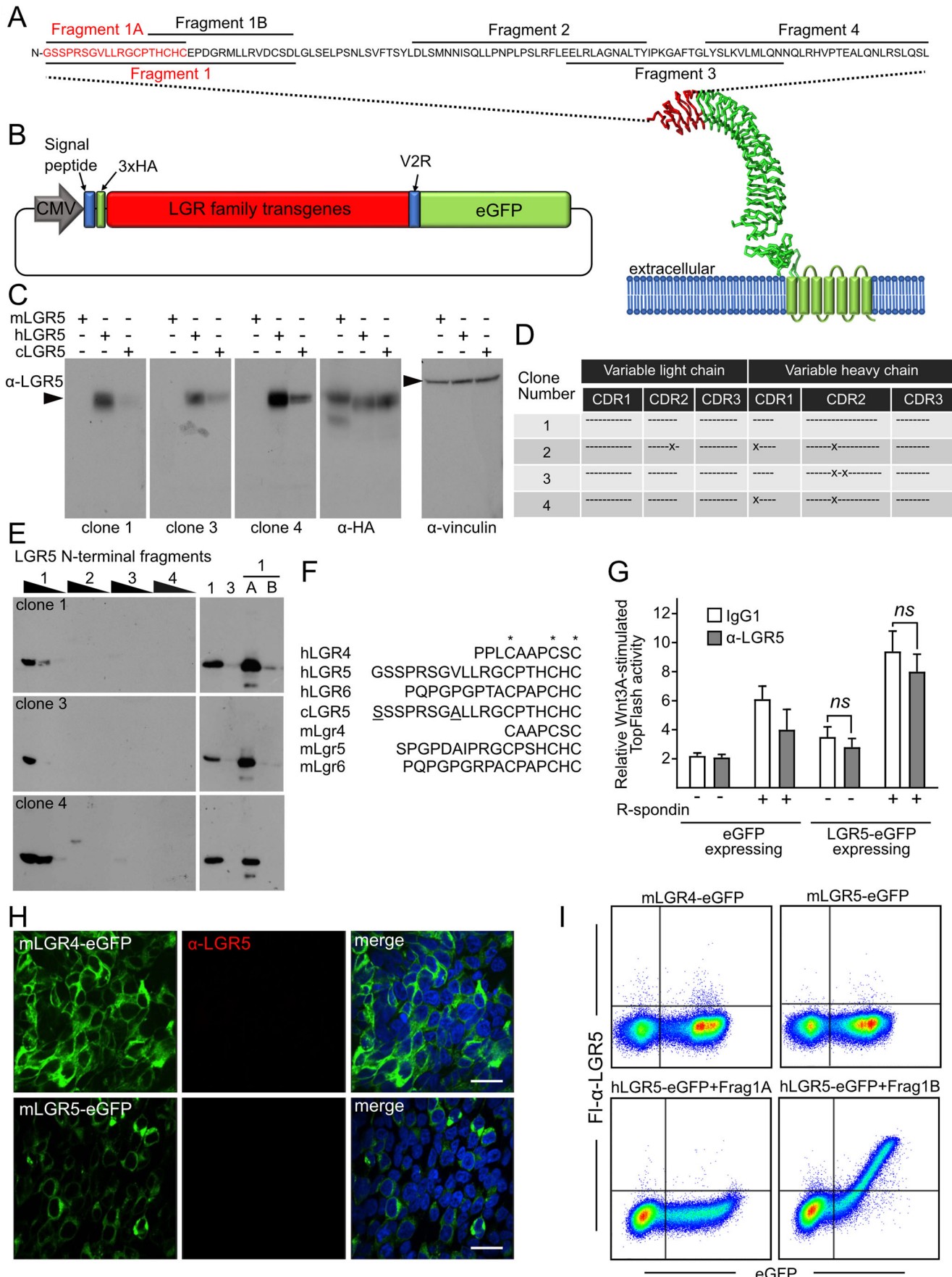

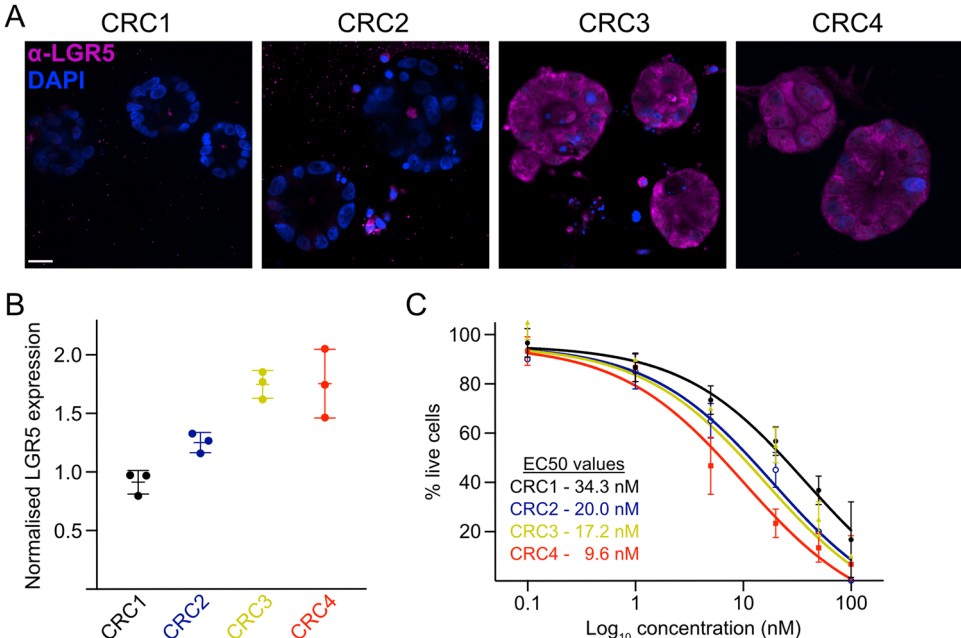

**Figure EV2. Sensitivity of CRC organoid models expressing variable LGR5 levels to α-LGR5v4-ADC treatment.**

(A) Immunofluorescent imaging of LGR5 in CRC organoid models. Images are representative of 2 independent experiments. Scale bar, 20 μm. (B) Relative *LGR5* transcript levels in the CRC organoids models measured by quantitative qRT-PCR with *TBP* as a reference gene. Data is presented as mean expression, $+/-$ SD for 3 biological replicates. (C) CRC organoid model killing with α-LGR5v4-ADC treatment quantified as the percent of CRC organoids at each treatment level that displayed more than 20% of cleaved caspase 3 positive component cells. Data for treatment of each organoid model is derived from a minimum of 10 datapoints at each concentration of α-LGR5v4-ADC with 2–3 independent biological replicates. Data is presented as mean expression, $+/-$ SD. Source data are available online for this figure.

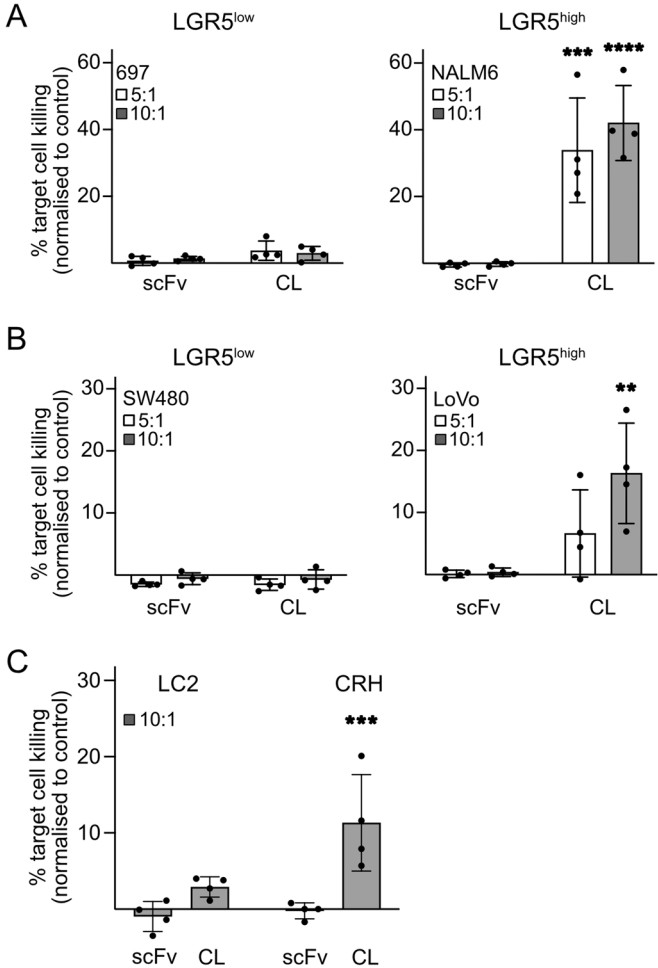

**Figure EV3. Differential sensitivity of pre-B-ALL and CRC cell lines as well as B-ALL patient samples to treatment with PBMCs and CL-BiTE.**

(A) LGR5low and LGR5high expressing human pre-B-ALL cell lines were incubated in the presence of cytotoxic CD8+ T cells with either scFv control or CL-BiTEs. Killing was assessed after 6 h at effector to target cell ratios of 5:1 and 10:1. Data shown is from one experiment using CD8+ T cells isolated from four individual healthy donors and is presented as mean expression, +/− SD. (B) LGR5low and LGR5high expressing human CRC cell lines were incubated in the presence of cytotoxic CD8+ T cells with either scFv control or CL-BiTEs. Killing was assessed after 6 h at effector to target cell ratios of 5:1 and 10:1. Data shown is from one experiment using CD8+ T cells isolated from four individual healthy donors and is presented as mean expression, +/− SD. (C) LGR5low and LGR5high expressing human pre-B-ALL patient samples were incubated in the presence of cytotoxic CD8+ T cells with either scFv control or CL-BiTEs. Killing was assessed after 9 h at an effector to target cell ratio of 10:1. Data shown is from one experiment using CD8+ T cells isolated from four individual healthy donors and error bars represent mean +/− SD. Source data are available online for this figure.

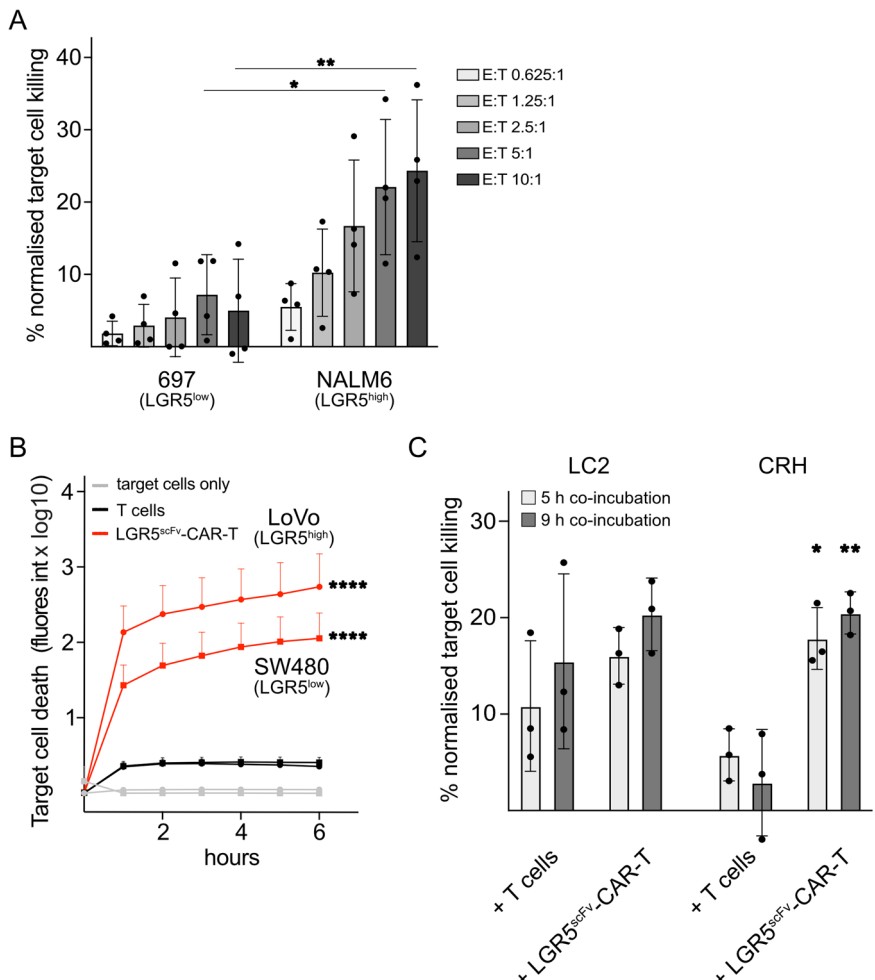

**Figure EV4. Differential sensitivity of pre-B-ALL and CRC cell lines as well as pre-B-ALL patient samples to treatment with LGR5scFV-CAR-T cells.**

(A) Increased killing of LGR5high preB-ALL NALM6 cells relative to LGR5low 697 cells by LGR5scFV-CAR-T cells. Data shown is from two experiments using LGR5scFV-CAR-T cells generated from a total of four independent healthy donors and is presented as mean expression, +/− SD. (B) Reduced sensitivity to LGR5scFV-CAR-T-cell killing of LGR5low CRC SW480 cells relative to LGR5high LoVo cells. Data shown is from two experiments using LGR5scFV-CAR-T cells generated from a total of three independent healthy donors and is presented as mean expression, +/− SEM. (C) The LGR5high CRH patient-derived pre-B-ALL cell model is more sensitive to killing by LGR5scFV-CAR-T cells compared to LGR5low pre-B-ALL LC2 patient cells. Data shown is from one experiment using LGR5scFV-CAR-T cells generated from a total of three independent healthy donors and is presented as mean expression, +/− SD. Source data are available online for this figure.

