## [Peer Review File · EMBO Molecular Medicine]

Novel immunotherapeutics against LGR5 to target multiple cancer types

Maike de la Roche, Marc de la Roche, Hung-Chang Chen, Nico Mueller, Katherine Stott, Chrysa Kapeni, Eilidh Rivers, Carolin Sauer, Flavio Beke, Stephen Walsh, Louise O'Brien, Amir Rafati Fard, Arman Ghodsinia, Olivier Giger, Inti Zlobec, Ioana Olan, Sarah Aitken, Matthew Hoare, Richard Mair, Eva Serrao, James Brenton, Alicia Garcia-Gimenez, Simon Richardson, Brian Huntly, David Spring, Mikkel-Ole Skjoedt, Karsten Skjodt, Nicola Ashman, Changtai Li, and Fadwa Joud

Corresponding authors: Maike de la Roche (maike.delarochec@cr.uk.cam.ac.uk) , Marc de la Roche (mad58@cam.ac.uk)

Review Timeline:

Submission Date:	16th Feb 24
Editorial Decision:	14th Mar 24
Revision Received:	8th Jul 24
Editorial Decision:	24th Jul 24
Revision Received:	30th Jul 24
Accepted:	1st Aug 24

Editor: Lise Roth

Transaction Report:

14th Mar 2024

Dear Dr. de la Roche,

Thank you for the submission of your manuscript to EMBO Molecular Medicine. We have now received feedback from the three reviewers who agreed to evaluate your manuscript. As you will see from the reports below, the referees acknowledge the interest of the study and are overall supporting publication of your work pending appropriate revisions.

In their reports, the referees mention that the main limitation of the work is the lack of safety studies due to the absence of cross-reactivity with muLGR5 and suggest different ways to address this point. We further consulted with them and agreed that generation of a humanized LGR5 mouse or experiments in non-human primates might be beyond the scope of the study. Therefore, we would ask you to perform *in vitro* killing experiments, as suggested by referee #3, and to discuss this limitation of your work in the manuscript.

Adequately addressing this point and other reviewers' concerns will be necessary for further considering the manuscript in our journal, and acceptance of the manuscript will entail a second round of review. EMBO Molecular Medicine encourages a single round of revision only and therefore, acceptance or rejection of the manuscript will depend on the completeness of your responses included in the next, final version of the manuscript. For this reason, and to save you from any frustrations in the end, I would strongly advise against returning an incomplete revision.

We are expecting your revised manuscript within three months, if you anticipate any delay, please contact us.

We require:

4) A .docx formatted letter INCLUDING the reviewers' reports and your detailed point-by-point responses to their comments. As part of the EMBO Press transparent editorial process, the point-by-point response is part of the Review Process File (RPF), which will be published alongside your paper.

5) A complete author checklist, which you can download from our author guidelines (<https://www.embopress.org/page/journal/17574684/authorguide#submissionofrevisions>). Please insert information in the checklist that is also reflected in the manuscript. The completed author checklist will also be part of the RPF.

6) It is mandatory to include a 'Data Availability' section after the Materials and Methods. Before submitting your revision, primary datasets produced in this study need to be deposited in an appropriate public database, and the accession numbers and database listed under 'Data Availability'. Please remember to provide a reviewer password if the datasets are not yet public (see <https://www.embopress.org/page/journal/17574684/authorguide#dataavailability>).

7) For data quantification: please specify the name of the statistical test used to generate error bars and P values, the number (n) of independent experiments (specify technical or biological replicates) underlying each data point and the test used to calculate p-values in each figure legend. The figure legends should contain a basic description of n, P and the test applied. Graphs must include a description of the bars and the error bars (s.d., s.e.m.). Please provide exact p values.

8) Our journal encourages inclusion of *data citations in the reference list* to directly cite datasets that were re-used and

obtained from public databases. Data citations in the article text are distinct from normal bibliographical citations and should directly link to the database records from which the data can be accessed. In the main text, data citations are formatted as follows: "Data ref: Smith et al, 2001" or "Data ref: NCBI Sequence Read Archive PRJNA342805, 2017". In the Reference list, data citations must be labeled with "[DATASET]". A data reference must provide the database name, accession number/identifiers and a resolvable link to the landing page from which the data can be accessed at the end of the reference. Further instructions are available at .

9) We replaced Supplementary Information with Expanded View (EV) Figures and Tables that are collapsible/expandable online. A maximum of 5 EV Figures can be typeset. EV Figures should be cited as 'Figure EV1, Figure EV2' etc... in the text and their respective legends should be included in the main text after the legends of regular figures.

10) The paper explained: EMBO Molecular Medicine articles are accompanied by a summary of the articles to emphasize the major findings in the paper and their medical implications for the non-specialist reader. Please provide a draft summary of your article highlighting

11) For more information: There is space at the end of each article to list relevant web links for further consultation by our readers. Could you identify some relevant ones and provide such information as well? Some examples are patient associations, relevant databases, OMIM/proteins/genes links, author's websites, etc...

12) Author contributions: CRediT has replaced the traditional author contributions section because it offers a systematic machine readable author contributions format that allows for more effective research assessment. Please remove the Authors Contributions from the manuscript and use the free text boxes beneath each contributing author's name in our system to add specific details on the author's contribution. More information is available in our guide to authors.

13) Disclosure statement and competing interests: We updated our journal's competing interests policy in January 2022 and request authors to consider both actual and perceived competing interests. Please review the policy <https://www.embopress.org/competing-interests> and update your competing interests if necessary.

14) Every published paper now includes a 'Synopsis' to further enhance discoverability. Synopses are displayed on the journal webpage and are freely accessible to all readers. They include a short stand first (maximum of 300 characters, including space) as well as 2-5 one-sentences bullet points that summarizes the paper. Please write the bullet points to summarize the key NEW findings. They should be designed to be complementary to the abstract - i.e. not repeat the same text. We encourage inclusion of key acronyms and quantitative information (maximum of 30 words / bullet point). Please use the passive voice. Please attach these in a separate file or send them by email, we will incorporate them accordingly.

15) As part of the EMBO Publications transparent editorial process initiative (see our Editorial at <http://embomolmed.embopress.org/content/2/9/329>), EMBO Molecular Medicine will publish online a Review Process File (RPF) to accompany accepted manuscripts.

In the event of acceptance, this file will be published in conjunction with your paper and will include the anonymous referee reports, your point-by-point response and all pertinent correspondence relating to the manuscript. Let us know whether you agree with the publication of the RPF and as here, if you want to remove or not any figures from it prior to publication. Please note that the Authors checklist will be published at the end of the RPF.

I look forward to receiving your revised manuscript.

Yours sincerely,

Lise Roth

***** Reviewer's comments *****

Referee #1 (Comments on Novelty/Model System for Author):

lack of use of mouse cross-reactive antibodies do not allow conclusions on the safety of the described agents and their therapeutic use/index

Referee #1 (Remarks for Author):

In their paper the authors describe novel and unique antibodies recognizing the N-term stretch of LGR5, that do not compete with R-Spondin-1, and validate LGR5 expression on the cell surface and LGR5 as a target for antibody drug conjugates, T cell engagers and CAR-T cells. The work presented is technically sound and comprehensive including suitable controls, the generally limited number of suitable, functional LGR5 antibodies makes the findings of this study of interest. I suggest the following points to be addressed

- The LGR5 antibodies recognize a linear epitope at the N-term as shown by WB, peptide binding etc, and LGR5 in its native state, discuss what is known from a structural point of view, also including prediction of structural elements, and how this compares to known antibodies from the literature/clinic
- For affinity determination ideally monovalent Fabs would be applied, how do affinities between protein and cell and the peptide stretch/fragment compare
- Can you comment on LGR5 expression on B cells on transcript and protein level given expression in Nalm-6 ALL cells
- The findings with ADCs, BiTEs and CAR-Ts are well performed and establish the efficacy of these agents and of targeting LGR5, the limitation of these findings (as mentioned in the discussion) given the expression of LGR5 on normal stem cells/tissues is that these studies do not show that LGR5 can be safely targeted in vivo with these approaches due to lack of cross-reactivity with muLGR5 which is a key requirement when thinking about therapeutic application. In this sense the use of ADCs may still be feasible due to the higher therapeutic index of ADCs as compared to more potent BiTE or CAR-T cell therapies. Do the authors have data for a muLGR5 surrogate antibody that could demonstrate safety
- The authors frequently refer to other LGR5 antibodies utilized, can you provide further information on the domain mapping of your antibody as compared to those antibodies, generally, it would have been good to include some of these antibodies for limited benchmarking experiments to show the equivalence or superiority of this novel antibody and its epitope

Referee #2 (Comments on Novelty/Model System for Author):

The authors generate new monoclonal antibodies that are specific for the N-terminus of human and cynomolgus LGR5. The characterization of the antibodies and the technical quality of the work is well done. The novelty is somewhat diminished by two other papers cited by the authors and describing anti LGR5 antibodies and the generation of antibody drug conjugates to target tumors that overexpress LGR5. The mAbs described here are of higher affinity and the authors extend the work to bispecific T cell engagers and chimeric antigen receptors. A limitation is that the safety of each of these modalities cannot be established in the NSG mouse models since the mAbs do not cross react with mouse LGR5. The mAbs do recognize cynomolgus LGR5 providing a real opportunity for future safety studies in non-human primates.

Referee #2 (Remarks for Author):

The authors describe the generation, characterization, and utility of monoclonal antibodies specific for human and cynomolgus macaque LGR5. The work is technically very nice and the antibodies are likely to find utility in future studies of LGR5 biology in

normal and transformed cells, and are of potential therapeutic interest.

Minor Points

1. There is some inconsistency in Figure 1 and supplementary Figure 1 in what antibody clones are used in western blot experiments. In SFigure 1E for example it says clones 1,3, and 4 were used in the text but the figure shows only one blot.
2. The data in Figure 2 shows high levels of LGR5 RNA expression in 32 of 35 ALL cases, yet flow cytometry is only shown for a few cell lines. Given the authors own demonstration that RNA levels do not always coincide with protein levels (ovarian cancer for example), showing flow cytometry data on patient ALL samples would strengthen the relevance.
3. The internalization of LGR5 after mAb binding is important for the application of ADC efficacy. Internalization after ligand binding has been shown to be important for LGR5 signaling, is there any evidence that naked mAb binding results in LGR5 signaling?
4. The in vivo experiments with ADCs, BiTEs, and CARs against Nalm-6 show transient antitumor activity but not complete tumor eradication. Given the heterogeneity of LGFR on tumors, it would be important to understand if the persisting (or regrowing) tumor cells continue to express LGR5.
5. A weakness in the paper is the inability to evaluate toxicity of the therapeutic modalities using the mAbs because they do not recognize murine LGR5. The authors should discuss the potential to evaluate safety of each modality in non-human primates given that the mAbs cross react with cynomolgus LGR5

Referee #3 (Comments on Novelty/Model System for Author):

Technical quality is generally very high except for the data in Fig 7B which has such big error bars it is very unlikely to be significantly different but this is not reflected in the main text.

Novelty is slightly lowered as 2 other LGR5 Abs have already been published and used to generate ADCs. However the generation of LGR5-BiTEs and LGR5scFv-CAR-T cells is novel and an exciting advance for these approaches, with robust efficacy/validation in vitro and in vivo.

Referee #3 (Remarks for Author):

The authors have generated an LGR5 specific Ab. The experiments to validate this are generally good and robust and highlight LGR5 is upregulated in several cancers and a potential attractive target for therapy. However two Lgr5 specific Abs have previously been published and used to generate LGR5-ADCs which show good efficacy in vivo so it is not clear if this is a major step forward from those papers (Junttila et al. *Sci Transl Med.* 2015;7(314):1-12. Gong et al *Mol Cancer Ther.* 2016;15(7):1580-90.). However the new LGR5 Ab is very versatile, and the generation of LGR5-BiTEs and LGR5scFv-CAR-T cells is novel and an exciting advance for these approaches, with robust efficacy/validation in vitro and in vivo.

The new approaches to target LGR5+ cancer cells show very good anti-tumour efficacy, and the experiments are all performed with the appropriate controls and presented well. However there remains a major question regarding the true benefit of these approaches given Lgr5 is expressed in normal stem cells. The new LGR5 Ab does not bind to mouse Lgr5 and therefore no in vivo toxicity assays can be performed which would greatly enhance the impact of this paper (please see specific comments below). Even so this is an exciting step forward in the potential to target LGR5 pharmacologically.

Specific points:

1. One sentence summary: First line '...the cancer cell marker LGR5..' Lgr5 is not a specific marker of cancer cells, please can the authors correct this.
2. Abstract: Authors says α -LGR5 stratifies Lgr5 overexpression in CRC, HCC and ALL cases - what do they mean by cases? Please be more specific here. By stratifies do the authors mean detects? Stratifies sounds like the wrong term here.
3. Abstract: 'Treatment 65 of the human pre-B-ALL tumours with two doses of α -LGR5-BiTE and human T cells over the course of an 11-day experiment led to a 2-fold reduction in tumour burden relative to control.' Which human pre-B-ALL tumours does this sentence refer to? Please be more precise explaining these experiments - this sounds like a clinical trial has been performed.
4. The introduction correctly describes that Lgr5 is a marker of stem cells in several tissues. Therefore a critical question to address when targeting Lgr5+ cancer cells, is the effect on the Lgr5+ stem cells. As the new α -LGR5 does not recognise mouse Lgr5, the toxicity is not able to be determined in vivo. A humanised LGR5 mouse will need to be generated which the α -LGR5 is able to recognise to facilitate investigating this important question which I expect is beyond the scope of a revision, but would be great if the authors are in a position to do it? If not, then some investigation of LGR5-ADC killing non-tumours epithelial cells in vitro is required please.
5. Sup Fig 1. Please include what h, m and c stand for in the fig legend.

6. Sup fig 2 shows Lgr5 expression is lower in healthy tissue compared to tumours but can α -LGR5 still bind to Lgr5+ cells in healthy tissue? This is an important question for the ADC regarding toxicity. Can the authors please perform some experiments to investigate this in living and fixed cells from the tissues with highest LGR5 expression from non-tumour tissue?

7. Fig 7 A and B labels are missing.

8. Fig 7B: Can the authors please indicate the exact assay that has been performed in Fig 7B. There is no significant statistical difference in the death of any of the cancer cell lines used between the T cell incubated and LGR5SCFV-CAR-T cell incubated experiments, however the main text and figure legend suggests there is. The IncuCyte data in Sup Fig 7B shows robust killing of cancer cells from LGR5SCFV-CAR-T cell incubation, so can the authors explain this set of data more clearly please? If there are differences in killing in different assays please provide some explanation as to why. The data in Supp Fig 7B might be better as a main figure?

Author response to referee comments:

We thank all referees for their insightful comments and helpful corrections that have enabled us to substantially improve the manuscript and more clearly showcase the experimental and therapeutic applications of the α -LGR5 immunotherapeutics that we have developed.

The general comment that all three referees touched on was our inability to assess safety for the three therapeutic modalities we developed in murine models owing to the fact that the α -LGR5 antibodies do not cross-react with murine LGR5. Under request from the Editor and referees, we have carried out extensive *in vitro* assays with the ADC and BiTE molecules as well as the α -LGR5^{scFv}-CAR-T cells to establish the therapeutic window for these α -LGR5-based immunotherapeutics for targeting cancer cells and healthy tissues, using both cell lines and primary human cell and organoid lines, as follows:

For the α -LGR5v4-ADC molecule, we have now carried out killing assays using 4 patient derived organoid CRC models from our biobanks. **New Figure EV2** contains new data that introduces 4 CRC models that display a range of LGR5 expression levels, determined using immunofluorescence with α -LGR5 as well as qRT-PCR to quantify transcript levels. LGR5 protein and transcript levels show very good correlation between the 4 organoid models. Moreover, the relative LGR5 expression levels correlate perfectly with sensitivity of the organoid models to α -LGR5-ADC treatment. The lowest LGR5-expressing organoid model, CRC-1, is the least sensitive to α -LGR5-ADC treatment with the highest EC50 values, whereas the highest LGR5 expressor, CRC4, is the most sensitive for α -LGR5-ADC treatment (lowest EC50 values). We note that the relative LGR5 expression levels in CRC1, determined through immunofluorescence, are higher than what we observe for healthy colon epithelial tissue in our TMA work (for instance in **Fig. 2A** and **Appendix Figure S1C**). The data supports a wide therapeutic window for α -LGR5v4-ADC treatment.

We have also generated similar data for α -LGR5-BiTE, in this instance using patient-derived pre-B-ALL samples as targets. We have opted not to use CRC models in these assays owing to insufficient penetration of the 3-dimensional organoid support substratum by BiTE-redirected T cells. The new data consists of cancer cell killing assays using the patient-derived pre-B-ALL samples, CRH and LC2. We have assessed LGR5 transcript and protein levels and confirmed high LGR5 expression in the CRH sample and low LGR5 expression by LC2 and SQ1 samples (**new Fig. 2H, I**). Importantly, treatment with α -LGR5-BiTE indicates that the high expressing CRH cells are particularly vulnerable, whereas the low LGR5 expressing LC2 patient sample is unaffected (**new Fig. EV3C**). We observe similar differential killing comparing low LGR5 expressing 697 and high expressing NALM6 B-ALL cell lines (**new Fig. EV3A**) and the low LGR5 expressing SW480 and high expressing LoVo CRC cell lines (**new Fig. EV3B**).

For the α -LGR5-CAR-T cell modality, we included new additional data showing efficacy of α -LGR5-CAR-T cells targeting the patient-derived pre-B-ALL samples. Our data demonstrate potent cell killing activity towards the high LGR5 expressing CRH samples and low level killing of the low LGR5 expressing LC2 sample (**new Fig. EV4C**). The data provides an excellent complement to the highly potent α -LGR5-CAR-T cell killing of the LGR5^{high} NALM6 and LoVo cell lines versus the refractory LGR5^{low} 697 and SW480 cell lines (**new Fig. EV4A and B**).

The second direct request from the Editor was to discuss limitations of the work and we have therefore included an additional paragraph within the Discussion.

We address the specific referee comments as follows:

Referee 1.

-The LGR5 antibodies recognize a linear epitope at the N-term as shown by WB, peptide binding etc, and LGR5 in its native state, discuss what is known from a structural point of view, also including prediction of structural elements, and how this compares to known antibodies from the literature/clinic

We have incorporated a discussion of structural elements of LGR5 and the target epitopes of α -LGR5 and other reported α -LGR5 antibodies in the Discussion section. To this end, we have incorporated **new Appendix Figure S7** that shows the location of the individual target epitopes on the LGR5 structure in the context of bound R-spondin.

In the discussion we have now added text (lines 804-815) explaining structural considerations for the antibodies referring to **Table 1** and the comparison to reported α -LGR5 antibodies in the table shown in **Appendix Table S1**.

-For affinity determination ideally monovalent Fabs would be applied, how do affinities between protein and cell and the peptide stretch/fragment compare

We have included text in the Discussion (lines 816-820) indicating that binding studies were carried out with the antibody moieties and the scFv fragment to enable direct comparison with literature values. We have used bio-layer interferometry (BLI; Octet platform) for highly accurate, direct measurements of antibody-epitope binding affinities, that are reported alongside binding affinities for the other reported antibodies in **Table 1**. The referee notes that in some cases (for MCLA-158 and 8F2) values are apparent Kd values, calculated using cell binding as readout, and may not be as accurate as the direct BLI measurement we and the BNC101 study carried out. We have therefore made note of the studies that report apparent Kd values through cell binding assays in **Appendix Table S1** to allow the reader to assess the accuracy of the data.

-Can you comment on LGR5 expression on B cells on transcript and protein level given expression in Nalm-6 ALL cells

We have now added text to compare LGR5 transcript levels in primary human B cells with expression levels in NALM6 ALL cells (lines 327-331).

-The findings with ADCs, BiTEs and CAR-Ts are well performed and establish the efficacy of these agents and of targeting LGR5, the limitation of these findings (as mentioned in the discussion) given the expression of LGR5 on normal stem cells/tissues is that these studies do not show that LGR5 can be safely targeted in vivo with these approaches due to lack of cross-reactivity with muLGR5 which is a key requirement when thinking about therapeutic application. In this sense the use of ADCs may still be feasible due to the higher therapeutic index of ADCs as compared to more potent BiTE or CAR-T cell therapies. Do the authors have data for a muLGR5 surrogate antibody that could demonstrate safety

Please see general response to editor and referees comments, above.

-The authors frequently refer to other LGR5 antibodies utilized, can you provide further information on the domain mapping of your antibody as compared to those antibodies, generally, it would have been good to include some of these antibodies for limited benchmarking experiments to show the equivalence or superiority of this novel antibody and its epitope

See new **Appendix Figure S7**, referred to above, where we have included the domain mapping data for α -LGR5 and other reported α -LGR5 antibodies.

We have decided to not include studies of other antibodies owing to the breadth of additional data that would have to be included in the study that includes expression, QC, *in vitro* and possibly *in vivo* experiments; we do not think this substantial additional data would advance knowledge of experimental and therapeutic deployment of α -LGR5.

Referee 2

For our response to general comment about safety studies, please refer to general response to referees and editor, above.

Minor points

1. *There is some inconsistency in Figure 1 and supplementary Figure 1 in what antibody clones are used in western blot experiments. In SFigure 1E for example it says clones 1,3, and 4 were used in the text but the figure shows only one blot. We have corrected the inconsistency in **Suppl. Fig. 1E, now Fig. EV1E** and updated figure and figure legend. We added a note in the text explaining why we proceeded with α -LGR5 clone 2 in subsequent studies (lines 150-157).*

2. *The data in Figure 2 shows high levels of LGR5 RNA expression in 32 of 35 ALL cases, yet flow cytometry is only shown for a few cell lines. Given the authors own demonstration that RNA levels do not always coincide with protein levels (ovarian cancer for example), showing flow cytometry data on patient ALL samples would strengthen the relevance.*

We have carried out additional flowcytometric analysis of LGR5 protein expression in three patient-derived pre-B-ALL lines. Interestingly, we find a good correlation between protein and transcript levels and have included this data as **new Figure 2I**. We included corresponding text in the results (lines 327-331) and discussion sections (lines 833-835).

3. *The internalization of LGR5 after mAb binding is important for the application of ADC efficacy. Internalization after ligand binding has been shown to be important for LGR5 signaling, is there any evidence that naked mAb binding results in LGR5 signaling?*

We have shown that α -LGR5 binding to LGR5 does not interfere with its binding to R-spondin and downstream potentiation of Wnt pathway activity (**Figure EV1G**).

We observe an identical pattern between fluorescent α -LGR5 internalisation (LoVo cells (**Appendix Figure S3C**), NALM6 cells (**Fig. 4B**) and HEK293T cells overexpressing LGR5 (**Fig. 4A**) and our LGR5 immunofluorescent data (LoVo cells (**Fig. 3H**), NALM6 cells (**Fig. 3C**), HEK293T cells overexpressing LGR5 (**Fig. 1C**)) suggesting that binding of α -LGR5 to LGR5 does not itself trigger receptor internalisation which is instead constitutive. Our data is consistent with published work from Gong et al (*Mol. Cell Biol.* 2012) that finds constitutive internalisation of LGR5 that is independent from antibody binding.

4. *The in vivo experiments with ADCs, BiTEs, and CARs against Nalm-6 show transient antitumor activity but not complete tumor eradication. Given the heterogeneity of LGFR on tumors, it would be important to understand if the persisting (or regrowing) tumor cells continue to express LGR5.*

This is an excellent observation and we have now developed a data set for the study. We have looked at residual tumours from the α -LGR5-BiTE study which retained sufficient tumour load in the spleen for analysis of LGR5 expression in NALM6 tumour cells. We used the CD20 probe to discriminate NALM6 cells and α -LGR5 to detect LGR5 expression and found overlapping signals. We did observe background with the antibody but were able to distinguish true positives by the staining intensity and pattern of cellular expression. Our analysis finds that all CD20 positive NALM6 cells express high levels of LGR5 in spleen sections from control PBMC treated or PBMC + α -LGR5-BiTE treated mice. We conclude that residual tumour load from treatment of tumour bearing mice with α -LGR5-based therapeutics are the consequence of incomplete tumour access and targeting and not due to downregulation of LGR5 expression in tumour cells. The new data is now included in the **new Figure 6F**.

5. *A weakness in the paper is the inability to evaluate toxicity of the therapeutic modalities using the mAbs because they do not recognize murine LGR5. The authors should discuss the potential to evaluate safety of each modality in non-human primates given that the mAbs cross react with cynomolgus LGR.*

For our response to general comment about safety studies, please refer to general response to referees and editor, above.

In the discussion (lines 905-906) we have added a paragraph and discussed the opportunity for toxicology studies in cynomolgus monkey.

Referee 3

For the referee's comment on safety studies, please refer to general response to referees and editor, above.

1. *One sentence summary: First line '...the cancer cell marker LGR5..' Lgr5 is not a specific marker of cancer cells, please can the authors correct this.*

We have corrected this.

2. *Abstract: Authors says α -LGR% stratifies Lgr5 overexpression in CRC, HCC and ALL cases - what do they mean by cases? Please be more specific here. By stratifies do the authors mean detects? Stratifies sounds like the wrong term here.*

We have replaced cases with biopsies and stratifies with detects in the Abstract.

3. *Abstract: 'Treatment 65 of the human pre-B-ALL tumours with two doses of α -LGR5-BiTE and human T cells over the course of an 11-day experiment led to a 2-fold reduction in tumour burden relative to control.' Which human pre-B-ALL tumours does this sentence refer to? Please be more precise explaining these experiments - this sounds like a clinical trial has been performed.*

We have altered the abstract and incorporated the requested change.

4. *The introduction correctly describes that Lgr5 is a marker of stem cells in several tissues. Therefore, a critical question to address when targeting Lgr5+ cancer cells, is the effect on the Lgr5+ stem cells. As the new α -LGR5 does not recognise mouse Lgr5, the toxicity is not able to be determined in vivo. A humanised LGR5 mouse will need to be generated which the α -LGR5 is able to recognise to facilitate investigating this important question which I expect is beyond the scope of a revision, but would be great if the authors are in a position to do it? If not, then some investigation of LGR5-ADC killing non-tumours epithelial cells in vitro is required please.*

For the referee's comment on safety studies, please refer to general response to referees and editor, above.

5. *Sup Fig 1. Please include what h, m and c stand for in the fig legend.*

We have updated the legend to include definitions of the terms.

6. *Sup fig 2 shows Lgr5 expression is lower in healthy tissue compared to tumours but can α -LGR5 still bind to Lgr5+ cells in healthy tissue? This is an important question for the ADC regarding toxicity. Can the authors please perform some experiments to investigate this in living and fixed cells from the tissues with highest LGR5 expression from non-tumour tissue?*

We find that in rare cases, we are able to detect LGR5 protein expression by healthy tissue. For instance, we have identified two instances where we can observe LGR5 expression in healthy tissue – colon epithelia in **Fig. 2A and Appendix Figure S1C** and fallopian tube tissue in **Appendix Figure S1H**. We note that LGR5 protein expression within these cells is much lower than in corresponding tumour cells expressing LGR5.

Further, we have carried out α -LGR5-ADC targeting studies on colorectal cancer organoids that display different levels of LGR5 protein expression – **new Figure EV2**. Intriguingly, we find an excellent correlation between higher levels of LGR5 protein expression and vulnerability of the organoids to treatment with α -LGR5-ADC. Note, that LGR5 expression levels in CRC1, albeit lower than the other CRC organoid models, are higher than in colon epithelial tissue.

It is important to note that we have not included “healthy” colon epithelial organoids in these studies. Culture of colon epithelial organoids requires persistent activation of the Wnt pathway through supplement of media with Wnt3A ligand. Because LGR5 is a target gene of the Wnt pathway, its protein levels will be artificially raised, thereby introducing enhanced, artefactual vulnerability of the organoids to α -LGR5-ADC treatment.

We have also detailed a number of other studies that we have included to demonstrate safety available in the general response to referees and editor, above.

7. *Fig 7 A and B labels are missing.*

We have added the labels.

8. *Fig 7B: Can the authors please indicate the exact assay that has been performed in Fig 7B. There is no significant statistical difference in the death of any of the cancer cell lines used between the T cell incubated and LGR5SCFV-CAR-T cell incubated experiments,*

however the main text and figure legend suggests there is. The incuocyte data in Sup Fig 7B shows robust killing of cancer cells from LGR5SCFV-CAR-T cell incubation, so can the authors explain this set of data more clearly please? If there is differences in killing in different assays please provide some explanation as to why. The data in Supp Fig 7B might be better as a main figure?

We agree with the reviewer and have now included the incuocyte killing assays from original Suppl. Fig 7B into the main manuscript. We now show the killing kinetics up to 6 hours as a combined figure of three independent incuocyte experiments (using 3 different donors for the production of the CAR T cells). Our results are consistent with killing assays carried out over the same 6 hour timeframe for the LGR5⁺ target cell lines NALM6 (suspension), HepG2 (adherent), and LoVo (adherent).

24th Jul 2024

Dear Dr. de la Roche,

Thank you for submitting your revised study. We have now received the feedback from referees #1 and #3, and as you will see below, while referee #1 is satisfied with the revisions, referee #3 still raises one issue that has not been adequately addressed. This does not require extensive experiments, and we would like you to address this remaining point in a minor round of revisions. Additionally, please address the following editorial issues:

1/ Manuscript text:

- Please note that emails bounced for Fadwa Joud (Fadwa.Joud@cruk.cama.co.uk), Nicola Ashman (na504@cam.ac.uk) and Matthew Hoare (Matthew.Hoare@cruk.cam.ac.uk). Please also provide a statement that all authors are aware of and agree with the addition of new author Changtai Li.
- Accept previous changes and only keep in track changes mode any new modification.
- Please provide up to 5 keywords.
- The order of the manuscript sections should be: Abstract, Keywords, The Paper Explained, Introduction, Results, Discussion, Methods, Acknowledgements, Disclosure and competing interests statement, References, Figure legends, Tables and their legends, Expanded View Figure legends.
- Methods:
 - o All Materials and Methods need to be described in the main text using our 'Structured Methods' format, which is now required for all research articles. According to this format, the Methods section includes a Reagents and Tools Table (listing key reagents, experimental models, software and relevant equipment and including their sources and relevant identifiers) followed by a Methods and Protocols section describing the methods using a step-by-step protocol format. A downloadable template (.docx) for the Reagents and Tools Table can be found in our author guidelines: <https://www.embopress.org/page/journal/17574684/authorguide#structuredmethods>
 - o Antibodies: please provide dilutions/concentrations used.
 - o Patient samples: please include a statement that the experiments conformed to the principles set out in the WMA Declaration of Helsinki and the Department of Health and Human Services Belmont Report.
 - o Animals: please provide the age of the mice at the time of experiment.
 - o Statistical analysis: please include a statement on sample size and inclusion/exclusion criteria.
 - o Availability of published material: We note that you included the sentence: "The a-LGR5 antibodies are patented (PCT/GB2023/050512) and under clinical consideration and so there may be some limitations in the reagents that could be provided." My understanding is that the patent is filed (and all information on reagents and antibodies are provided in this manuscript in a transparent way), and the issue is not on intellectual property but rather on commercial distribution, is that correct? As per our policy on published material (<https://www.embopress.org/page/journal/17574684/authorguide#availabilityofpublishedmaterial>), "the authors agree to make available to colleagues in academic research all new reagents, including [...] antibodies, that were used in the research reported and that are not available from public repositories or commercial suppliers. [...] Materials must be made available at a reasonable cost that reflects production and distribution." We would therefore ask that you please remove this sentence from the manuscript. Please let us know if anything is unclear or if I misunderstood the issue.
 - Data Availability: In case you have no data that requires deposition in a public database, please state: "This study includes no data deposited in external repositories."
 - Acknowledgements should be merged with funding. The funding information provided in this section should match the information entered in the submission system (currently, CRUK CI PhD studentships to NM and FB and the NIHR Cambridge Biomedical Research Centre is not entered in the submission system).
 - "Competing interests" should be renamed "Disclosure statement and competing interests".
 - References: please reformat the references in alphabetical order with 10 authors listed before et al. DOIs should be removed for all published articles.

2/ Figures and Appendix:

- Please remove the figures from the manuscript text.
- Please make sure that all figures/figure panels are referenced in the text (currently, a callout is missing for Appendix Table S1; and there are callouts for a Suppl. Fig. 1A and Suppl. Fig. 2H, please correct).
- Appendix: please remove blue font.
- Please address the queries from our data editors in the figure legends (if not yet addressed):
 1. Please define the annotated p values ****/***/**/* as well as provide the exact p-values for the same in the legend of figure 2b, d, h; 4d-e; 5c, e-f, h; 6a-e; 7a-c, e; EV 3a-c; EV 4a-c; as appropriate.
 2. Please note that information related to n is missing in the legends of figures 2b, d-f; 7c; EV 4a-c.
 3. Please note that the error bars are not defined in the legends of figures 2b, d-f, h; EV 2c; EV 4a-c.
 4. Please note that the measure of center for the error bars needs to be defined in the legends of figures 3a, f, j; 4d-e; 5c, e-f, h; 6a-e; 7a-c, e; EV 1g; EV 2b; EV 3a-c.

5. Please note that the scale bar needs to be defined for figure EV 1h.
6. Please note that the white arrowheads are not defined in the legend of figure 2a. This needs to be rectified.

3/ Checklist:

- Please adjust the statement provided for "Newly Created Materials" (as discussed above).
- Please fill in "Experimental study design and statistics" / "sample size" and "inclusion/exclusion criteria".

4/ Please add the Paper Explained to the main manuscript file.

5/ As part of the EMBO Publications transparent editorial process initiative (see our Editorial at <http://embomolmed.embopress.org/content/2/9/329>), EMBO Molecular Medicine will publish online a Review Process File (RPF) to accompany accepted manuscripts.

This file will be published in conjunction with your paper and will include the anonymous referee reports, your point-by-point response and all pertinent correspondence relating to the manuscript. Let us know whether you agree with the publication of the RPF.

I look forward to receiving your revised manuscript.

With kind regards,

Lise Roth

***** Reviewer's comments *****

Referee #1 (Comments on Novelty/Model System for Author):

experiments are performed sound and well

Referee #1 (Remarks for Author):

revisions address the comments raised

Referee #3 (Comments on Novelty/Model System for Author):

The technical quality, novelty, impact and models used are all excellent.

Referee #3 (Remarks for Author):

I am happy the authors have addressed all of my minor points, however my major point of Lgr5-ADC toxicity still persists.

To address the question of Lgr5-ADC toxicity the authors have correlated that CRC organoids with increased Lgr5 protein levels are more sensitive to Lgr5-ADC treatment. They have also shown that protein levels of Lgr5 are lower in normal colon epithelium than colon cancer cells, and other tissues. However, these new data do not address the question of the response of normal colon epithelial cells to Lgr5-ADC treatment, either in vitro or in vivo. Although I think generating a humanised Lgr5 mouse is beyond the scope of a revision for this paper, it is not beyond the scope to treat some normal colon organoids with Lgr5-ADC. The authors indicate they did not perform this because the culture media needs Wnt3a, but normal colonic epithelium also requires Wnt ligands so this is still a valid experiment that should be included to help determine if the Lgr5-ADC is toxic to normal colon epithelium. If the colonic organoids die after Lgr5-ADC treatment this is not necessarily detrimental to this ADC as

a potential treatment, as the intestinal epithelium is able to re-populate after deleterious insults (eg genetic deletion of Myc or Fzd7) which is not observed ex vivo. This could then be investigated further in subsequent projects/publications using a humanised Lgr5 mouse and lineage tracing experiments.

Responses to Referees comments

We thank you for the referees' comments on the manuscript that we have addressed as follows:

Referee #1 (Comments on Novelty/Model System for Author):

experiments are performed sound and well

Referee #1 (Remarks for Author):

revisions address the comments raised

We thank Referee #1 for their input to the manuscript.

Referee #3 (Comments on Novelty/Model System for Author):

The technical quality, novelty, impact and models used are all excellent.

Referee #3 (Remarks for Author):

I am happy the authors have addressed all of my minor points, however my major point of Lgr5-ADC toxicity still persists.

To address the question of Lgr5-ADC toxicity the authors have correlated that CRC organoids with increased Lgr5 protein levels are more sensitive to Lgr5-ADC treatment. They have also shown that protein levels of Lgr5 are lower in normal colon epithelium then colon cancer

cells, and other tissues. However, these new data do not address the question of the response of normal colon epithelial cells to Lgr5-ADC treatment, either in vitro or in vivo. Although I think generating a humanised Lgr5 mouse is beyond the scope of a revision for this paper, it is not beyond the scope to treat some normal colon organoids with Lgr5-ADC. The authors indicate they did not perform this because the culture media needs Wnt3a, but normal colonic epithelium also requires Wnt ligands so this is still a valid experiment that should be included to help determine if the Lgr5-ADC is toxic to normal colon epithelium. If the colonic organoids die after Lgr5-ADC treatment this is not necessarily detrimental to this ADC as a potential treatment, as the intestinal epithelium is able to re-populate after deleterious insults (eg genetic deletion of Myc or Fzd7) which is not observed ex vivo. This could then be investigated further in subsequent projects/publications using a humanised Lgr5 mouse and lineage tracing experiments.

We thank Referee #3 for their input to the manuscript. Referee #3 makes the argument for testing the LGR5-ADC on healthy colon organoids. At face value this is a logical experiment to carry out. However, the main issue is that the healthy colon epithelial organoids will not grow without supplementation of the culture media with excess WNT ligands. We apologise that we were not detailed enough in our explanation of why the addition of WNT ligands to the culture media is disqualifying for this experiment: activation of the WNT signalling pathway will lead to expression of target genes. LGR5 is a WNT pathway target gene and so in normal colonic organoids LGR5 protein levels will be artificially high. Artificially high levels of LGR5 render faithful interpretation of healthy organoid susceptibility to LGR5-ADC treatment impossible.

We have now included text in the discussion explaining this.

Moreover, the reviewer also notes that regardless of the outcome of this line of investigation there will be scope for future study. We interpret this to mean that there will be no added value of this experiment to the study. We fully agree that further *in vivo* safety studies are an important extension to the work and we are currently exploring the means to do so in our follow-on study.

Maike de la Roche

Marc de la Roche

July 27th, 2024

1st Aug 2024

Dear Dr. de la Roche,

Thank you for submitting your revised files. I am pleased to inform you that your manuscript is accepted for publication and is now being sent to our publisher to be included in the next available issue of EMBO Molecular Medicine!

If you have any questions, please do not hesitate to contact the Editorial Office.

Congratulations on your nice work!

With kind regards,

Lise
